# Concatenated Matrix SVD: Compression Bounds, Incremental Approximation, and Error-Constrained Clustering

**Maksym Shamrai**                                                      *mshamrai@macpaw.com*
*Institute of Mathematics of NAS of Ukraine*
*MacPaw Research*

**Reviewed on OpenReview:** *https://openreview.net/forum?id=E9n35dehqx*

## Abstract

Large collections of matrices arise throughout modern machine learning, signal processing, and scientific computing, where they are commonly compressed by concatenation followed by truncated singular value decomposition (SVD). This strategy enables parameter sharing and efficient reconstruction and has been widely adopted across domains ranging from multi-view learning and signal processing to neural network compression. However, it leaves a fundamental question unanswered: *which matrices can be safely concatenated and compressed together under explicit reconstruction error constraints?* Existing approaches rely on heuristic or architecture-specific grouping and provide no principled guarantees on the resulting SVD approximation error. In the present work, we introduce a theory-driven framework for *compression-aware clustering* of matrices under SVD compression constraints. Our analysis establishes new spectral bounds for horizontally concatenated matrices, deriving global upper bounds on the optimal rank-$r$ SVD reconstruction error from lower bounds on singular value growth. The first bound follows from Weyl-type monotonicity under blockwise extensions, while the second leverages singular values of incremental residuals to yield tighter, per-block guarantees. We further develop an efficient approximate estimator based on incremental truncated SVD that tracks dominant singular values without forming the full concatenated matrix. Therefore, we propose three clustering algorithms that merge matrices only when their predicted joint SVD compression error remains below a user-specified threshold. The algorithms span a trade-off between speed, provable accuracy, and scalability, enabling compression-aware clustering with explicit error control.[1]

Large collections of matrices naturally arise in a wide range of applications, including multi-view representation learning, temporal aggregation of features, scientific simulations, neural network activations, and model parameter compression. A fundamental tool for compressing such data is the truncated Singular Value Decomposition (SVD), which provides an optimal low-rank approximation of a matrix in the Frobenius norm. By representing a matrix through a small number of dominant singular vectors, SVD enables compact storage, noise reduction, and efficient downstream computation. As a result, low-rank approximation via SVD is a standard building block in modern machine learning, signal processing, and scientific computing (Andrews & Patterson, 1976; Sun, 2013; Sarwar et al., 2000; Jolliffe, 2002).

When multiple matrices are available, a natural extension is to concatenate them (e.g., horizontally) and apply a single truncated SVD to the resulting matrix. This produces a shared low-rank representation across all blocks, enabling parameter sharing and coordinated compression. In particular, when matrices exhibit aligned or partially overlapping column spaces, a joint low-rank factorization can achieve substantially higher compression than compressing each matrix independently, as redundant directions are represented only once in the shared basis. For example, in neural network compression, weight matrices from different layers or components can be jointly factorized to share a common basis (Wang et al., 2025b; Lu et al., 2025; Wang et al., 2025a; Li et al., 2025); in scientific simulations, multiple snapshots or parameter instances can be

---

[1]Code is available online `https://github.com/mshamrai/concatenated-matrix-svd`.

compressed into a unified low-dimensional representation using snapshot-based methods (Sirovich, 1987); and in multi-view learning, features from different modalities can be embedded into a common latent space via joint low-rank representations (Hardoon et al., 2004; Kumar & Daumé, 2011). Compared to tensor decomposition methods such as Tucker or higher-order SVD (HOSVD) (Tucker, 1966; De Lathauwer et al., 2000b;a; Kolda & Bader, 2009), this matrix-based approach yields a single shared basis, avoids multilinear reconstruction, and integrates naturally with pipelines based on linear transformations. These advantages make concatenation followed by SVD a simple and effective strategy for compressing collections of matrices in practice.

However, the effectiveness of concatenated SVD critically depends on the alignment of the underlying subspaces. When matrices are misaligned, their concatenation can substantially increase the effective rank, and a shared low-rank approximation may incur significantly larger reconstruction error than compressing each matrix independently. For instance, two matrices that are individually low-rank but span different subspaces may produce a concatenated matrix with substantially higher rank. This leads to a fundamental question: *given a large collection of matrices, how can we efficiently determine which subsets should be concatenated and compressed together so that their joint low-rank approximation error remains below a prescribed tolerance?*

In practice, existing compression pipelines rely on heuristic grouping strategies, architectural constraints, or domain-specific assumptions to decide which matrices share a low-rank basis (Wang et al., 2025b; Lu et al., 2025; Li et al., 2025). Such approaches provide no explicit control over the resulting SVD reconstruction error and offer no guarantees that merging additional matrices will not violate accuracy requirements.

At a high level, the difficulty stems from the fact that concatenation fundamentally alters the singular value spectrum of the resulting matrix. While individual blocks may admit accurate low-rank approximations, their concatenation can increase effective rank and degrade approximation quality. Determining whether multiple matrices can safely share a low-rank representation therefore requires understanding how the singular values of the concatenated matrix evolve as new blocks are appended. This is inherently a spectral question and cannot be addressed by distance-based clustering or purely geometric similarity measures, as used in classical clustering methods.

In this work, we introduce the first theoretically grounded and computationally efficient framework for *compression-aware clustering* of matrices under explicit SVD reconstruction constraints. Our approach is based on new spectral bounds that characterize how singular values evolve under horizontal concatenation. Specifically, we derive two lower bounds on the singular values of concatenated matrices using classical matrix perturbation theory. The first bound follows from Weyl-type monotonicity under blockwise extensions (Weyl, 1912; Stewart & Sun, 1990), providing conservative but fast guarantees. The second, sharper bound exploits the singular values of incremental residuals, capturing the orthogonal contribution of each appended block and yielding tighter per-block estimates. Leveraging these spectral lower bounds, we derive corresponding global upper bounds on the optimal rank-$r$ SVD reconstruction error of the concatenated matrix. These bounds are exact in the non-truncated case and remain tight in practice under truncation. To enable scalability, we further develop an efficient approximate estimator based on incremental truncated SVD. This estimator maintains approximate dominant singular values as blocks are appended, drawing on ideas from incremental PCA and SVD (Hall et al., 1998; Levy & Lindenbaum, 1998; Brand, 2002) and randomized low-rank approximation (Halko et al., 2011; Tropp et al., 2017), while avoiding construction of the full concatenated matrix. Importantly, we do not claim a new incremental SVD algorithm, rather, we show how incremental singular value estimation can be repurposed as a decision-making primitive for compression-aware clustering under explicit error budgets.

Building on these theoretical results, we design three clustering algorithms that explicitly enforce SVD compression constraints. The first is a fast method based on Weyl-monotone bounds. The second provides provable reconstruction accuracy using residual-based spectral bounds. The third is a scalable approximate method that employs incremental truncated SVD to balance efficiency and approximation quality. All three algorithms merge matrices only when the predicted SVD reconstruction error of their concatenation lies below a user-specified threshold, thereby providing explicit and interpretable accuracy control.

In summary, this work establishes concatenation-aware SVD compression as a principled foundation for clustering and compressing large collections of matrices. By unifying spectral perturbation theory, incremental singular value estimation, and compression-driven clustering, we provide both theoretical guarantees and practical algorithms.

# 1 Related Work

**SVD-based compression of matrix collections.** Truncated singular value decomposition (SVD) is a fundamental tool for matrix compression, providing the optimal low-rank approximation of a matrix in the Frobenius norm (Eckart & Young, 1936; Mirsky, 1960; Jolliffe, 2002; Halko et al., 2011). By representing a matrix through a small number of dominant singular vectors, truncated SVD enables compact storage, noise reduction, and efficient downstream computation. A common and long-standing extension in many domains is to horizontally concatenate matrices and apply a single truncated SVD to the resulting matrix. This idea appears, for example, in principal component analysis applied to stacked data matrices (Jolliffe, 2002), as well as in the method of snapshots for model reduction (Sirovich, 1987). This yields a shared low-rank factorization across all blocks, enabling direct reconstruction of the original matrices without higher-order tensor contractions or complex decoding procedures.

Concatenated SVD has been successfully applied across a broad range of domains. In large language models, joint SVD of concatenated weight matrices has been used to share low-rank projections across attention components, layers, or experts, enabling parameter reduction while preserving accuracy. Representative examples include unified QKV decompositions (Wang et al., 2025b), intra-layer shared projections (Lu et al., 2025), cross-layer parameter sharing (Wang et al., 2025a), and expert merging in mixture-of-experts architectures (Li et al., 2025; Chaichana et al., 2025). In these settings, concatenation is typically guided by architectural structure (e.g., matrices belonging to the same layer or module) or semantic similarity (e.g., adjacent layers or related experts). Related ideas also appear in wireless signal processing, where concatenated SVD is used to design shared precoders across frequency bands (Zhang et al., 2016), as well as in neuroscience and genomics, where large collections of measurements are concatenated to obtain global low-dimensional representations across sessions, experimental conditions, or chromosomes (Nietz et al., 2023; Zhou et al., 2025). Across these application areas, concatenated SVD serves as a powerful tool for extracting shared structure from collections of matrices.

Despite its empirical success, existing uses of concatenated SVD rely on *predefined or heuristic grouping* of matrices. The decision of which matrices should share a low-rank basis is typically made manually based on domain knowledge, architectural constraints, or simple similarity measures. However, such empirical grouping strategies do not provide guarantees on the resulting reconstruction error. In particular, concatenating matrices based on heuristic similarity or architectural proximity can lead to failure cases: when matrices are not well aligned, their joint representation can exhibit substantially higher effective rank, resulting in significantly larger approximation error than compressing them independently. This makes the choice of which matrices to concatenate a critical component of the compression pipeline, rather than a purely heuristic design decision. Consequently, one must determine *which subsets of matrices should be concatenated* so that joint compression improves parameter efficiency while still satisfying a prescribed reconstruction error or compression target. Crucially, existing approaches do not provide a principled mechanism for making this decision. As a result, compression quality depends heavily on ad hoc design choices, and there are no guarantees that merging additional matrices will not violate reconstruction constraints.

In contrast, the present work formulates matrix grouping as a *compression-driven clustering problem*. Rather than clustering matrices based on ambient-space distances or semantic heuristics, cluster formation is governed directly by predicted low-rank approximation error of the concatenated matrix. This enables principled selection of matrix groups under explicit reconstruction error budgets.

**Incremental low-rank approximation.** Incremental estimation of dominant singular values and singular subspaces has a long history in numerical linear algebra and machine learning. Early work on incremental principal component analysis and online SVD (Hall et al., 1998; Levy & Lindenbaum, 1998) describes how to update covariance eigenbases as new samples arrive. Brand's incremental SVD algorithms (Brand, 2002;

2006) extend these ideas to rank-one and block updates of the thin SVD, directly covering the case of appending new columns, which is equivalent to horizontal concatenation. Streaming PCA and subspace tracking methods (Warmuth & Kuzmin, 2008; Mitliagkas et al., 2013) maintain approximate dominant invariant subspaces under stochastic or adversarial updates. Randomized low-rank approximation techniques (Halko et al., 2011; Woodruff, 2014) further improve scalability by maintaining approximate bases via random projections and periodic truncation.

The incremental truncated SVD estimator used in this work follows a classical design pattern and does not constitute a new SVD algorithm. Its update mechanism is equivalent to well-known incremental PCA and block-update SVD methods. The novelty of this work lies instead in how such estimators are used: we connect incremental singular value tracking to *explicit reconstruction-error control for concatenated matrices*, and embed it into clustering procedures whose merge decisions are driven directly by predicted low-rank approximation error. To our knowledge, existing incremental SVD and streaming PCA methods are not used to guide clustering or grouping under explicit SVD compression constraints.

## 2 Problem Formulation

Consider a collection of real matrices

$$\mathcal{A} = \{A_i\}_{i=1}^N, \qquad A_i \in \mathbb{R}^{m \times n_i}, \tag{1}$$

that must be stored and manipulated under a limited memory. Our goal consists of replacing the original matrices by compressed surrogates $\{\widehat{A}_i\}_{i=1}^N$ that preserve essential structure while employing as few real numbers as possible.

We measure fidelity by using the Frobenius norm

$$\mathcal{L}(\widehat{\mathcal{A}}, \mathcal{A}) = \left( \sum_{i=1}^N \|A_i - \widehat{A}_i(\Theta)\|_F^2 \right)^{1/2},$$

where a collection of real values $\Theta$ is parameterizing the compressed representation. Let $\mathrm{mem}(\Theta)$ denote the number of real values required to store $\Theta$. The *error-constrained memory minimization* principle reads as

$$\min_{\Theta} \ \mathrm{mem}(\Theta) \quad \text{s.t.} \quad \mathcal{L}(\widehat{\mathcal{A}}, \mathcal{A}) \leq \varepsilon, \tag{2}$$

where $\varepsilon > 0$ prescribes the maximal admissible distortion.

Compressing each matrix independently ignores potential *redundancy across the collection*. We assume that many matrices possess similar column spaces, which enables their joint representation via a shared low-rank basis. This motivates clustering followed by a joint factorization.

Recall that $A_i \in \mathbb{R}^{m \times n_i}$ for $i = 1, \ldots, N$, and let $\Pi = \{C_1, \ldots, C_K\}$ be a partition of this index set. For each cluster $C_c \in \Pi$, consider the concatenated matrix

$$M_c = [A_i]_{i \in C_c} \in \mathbb{R}^{m \times N_c}, \qquad \text{where} \quad N_c = \sum_{i \in C_c} n_i.$$

For each cluster $C_c$, let $r_c$ denote the target rank. We compute a rank-$r_c$ truncated SVD of $M_c$,

$$M_c \approx U_c S_c V_c^\top,$$

where $U_c \in \mathbb{R}^{m \times r_c}$ and $V_c \in \mathbb{R}^{N_c \times r_c}$ have orthonormal columns, and $S_c \in \mathbb{R}^{r_c \times r_c}$ contains the leading singular values. This provides a low-rank representation of the concatenated matrix $M_c$. Since the diagonal matrix $S_c$ can be absorbed into either factor of the decomposition, we keep only two matrices per cluster,

$$\widetilde{U}_c = U_c S_c \in \mathbb{R}^{m \times r_c}, \qquad V_c \in \mathbb{R}^{N_c \times r_c}.$$

This reduces storage while preserving a rank-$r_c$ representation. The memory footprint for cluster $C_c$ is therefore

$$\text{mem}_c = r_c(m + N_c),$$

consisting of $mr_c$ entries for $\widetilde{U}_c$ and $N_c r_c$ for $V_c$.

To reconstruct an approximation of each original matrix $A_i$, we split $V_c$ column-wise according to the block dimensions $n_i$. If $V_{c,i} \in \mathbb{R}^{n_i \times r_c}$ denotes the submatrix corresponding to $A_i$, then the recovered approximation is given by

$$\widehat{A}_i = \widetilde{U}_c V_{c,i}^\top, \qquad i \in C_c.$$

Thus each $A_i$ is represented using only the shared left basis $\widetilde{U}_c$ and its cluster-specific coefficient block $V_{c,i}$.

Because the rank-$r_c$ truncated SVD is the optimal Frobenius norm approximation of $M_c$ by the Eckart-Young-Mirsky theorem (see Appendix A for the preliminaries), the approximation error for cluster $C_c$ is precisely the energy in the discarded singular values:

$$\mathcal{L}_c = \left( \sum_{i \in C_c} \|A_i - \widehat{A}_i\|_F^2 \right)^{1/2} = \left( \sum_{j > r_c} \sigma_j^2(M_c) \right)^{1/2},$$

where $\sigma_j(M_c)$ denotes the $j$-th largest singular value of $M_c$, with singular values ordered non-increasingly.

The error-constrained memory minimization problem under cluster–concatenate–SVD representation with two stored matrices per cluster reads as

$$\min_{\Pi, \{r_c\}} \quad \sum_{c=1}^{K} r_c (m + N_c) \qquad \text{s.t.} \quad \left( \sum_{c=1}^{K} \sum_{j > r_c} \sigma_j^2(M_c) \right)^{1/2} \leq \varepsilon, \tag{3}$$

where $\Pi = \{C_1, \ldots, C_K\}$ is a partition of $\{1, \ldots, N\}$, $r_c$ is the rank assigned to cluster $C_c$, $m$ is the row dimension of all matrices, $N_c$ is the total number of columns in cluster $C_c$, and $\varepsilon > 0$ is the user-specified reconstruction tolerance controlling the achievable trade-off between accuracy and compression.

This formulation explicitly balances *storage per cluster* and *approximation error*. Adjusting $\varepsilon$ or the cluster-wise ranks $r_c$ tunes the balance between memory footprint and approximation quality.

**Why not optimize equation 3 directly?** While the formulation in equation 3 is compact and conceptually appealing, it is important to clarify why our approach does *not* attempt to solve it directly. The difficulty is twofold. First, the optimization over the partition $\Pi$ is inherently *combinatorial*. Even for fixed ranks $\{r_c\}$, determining an optimal clustering that minimizes the aggregate spectral tail $\sum_c \sum_{j > r_c} \sigma_j^2(M_c)$ subsumes hard partitioning problems and is NP-hard in general. Thus, global optimization over $\Pi$ is computationally intractable beyond very small instances. Second, even evaluating the objective is expensive. For a candidate cluster $C_c$, computing $\sigma_j(M_c)$ requires forming the concatenated matrix and performing at least a partial SVD, which scales with the total column dimension $N_c$. Embedding such spectral computations inside a combinatorial search over partitions is prohibitive in both time and memory.

For these reasons, our goal is not global optimality of equation 3, but rather the design of *provably safe local decisions* that guarantee feasibility of the error constraint. This perspective naturally leads to greedy clustering strategies driven by certified merge rules, rather than direct optimization. The key algorithmic primitive underlying our method is a *merge certificate* that allows one to decide whether two groups of matrices can be safely merged *without explicitly computing* the singular values of the concatenated matrix.

**Definition 1** (Compression-aware merge certificate). *Let $M = [M_1, M_2]$ denote the concatenation of two matrices. A compression-aware merge certificate is a computable condition $\mathcal{C}(M_1, M_2, r, \varepsilon)$ such that*

$$\mathcal{C}(M_1, M_2, r, \varepsilon) \implies \mathcal{E}_r(M) = \left( \sum_{j > r} \sigma_j^2(M) \right)^{1/2} \leq \varepsilon,$$

*without explicitly computing the singular values $\{\sigma_j(M)\}$.*

Such certificates enable greedy clustering schemes in which clusters are merged only when the resulting approximation error is *provably admissible.* Importantly, this shifts the computational burden from repeated large-scale SVDs to inexpensive spectral surrogates and bounds. This philosophy mirrors *safe screening rules* in sparse optimization, where variables are eliminated or grouped based on certificates that preserve optimality or feasibility, without solving the full problem (Ghaoui et al., 2010; Fercoq et al., 2015).

**Technical challenges.** The analysis of concatenated SVD presents several nontrivial challenges. First, the singular values of a concatenated matrix $M = [A_1, \ldots, A_K]$ do not decompose across blocks, as cross-block interactions in $M^\top M$ couple the spectra of individual matrices, preventing direct application of standard spectral results to individual blocks. Second, clustering decisions must be made without explicitly forming the full concatenated matrix, requiring error bounds that depend only on quantities that can be computed incrementally, such as projections onto the current subspace or block-wise statistics. Finally, the clustering problem is inherently combinatorial, since the number of possible partitions grows exponentially with the number of blocks, making exhaustive search infeasible and necessitating the use of efficiently computable merge criteria derived from such bounds. These challenges motivate the development of bounds that are both *computationally tractable* and *sufficiently tight* to guide clustering decisions in practice.

## 3 Methodology

We now develop our compression-aware clustering methodology. We begin with a very fast but coarse upper bound on the SVD compression error for concatenated matrices, derived from the Weyl monotonicity, and use it to construct a fast clustering algorithm. In the following subsections we refine this bound using the residuals and incremental SVD.

### 3.1 Weyl-Based Upper Bound for Concatenated SVD Compression

Our first result provides a simple *global* upper bound on the optimal rank-$r$ SVD compression error of a concatenated matrix in terms of the individual blocks. It relies only on the Frobenius norms and the leading singular values of the blocks and is therefore extremely cheap to evaluate. See the proof in Appendix C.1.

**Theorem 1** (Upper bound for SVD compression of a concatenated matrix). *Let $A_j \in \mathbb{R}^{m \times n_j}$ for $j = 1, \ldots, K$ and set*

$$M = \begin{bmatrix} A_1 & A_2 & \cdots & A_K \end{bmatrix} \in \mathbb{R}^{m \times (n_1 + \cdots + n_K)}.$$

*Denote by $\sigma_1(\cdot) \geq \sigma_2(\cdot) \geq \cdots$ the singular values of a matrix, and define the optimal rank-r approximation error in the Frobenius norm by*

$$\mathcal{E}_r^2(M) \; := \; \min_{\mathrm{rank}(X) \leq r} \|M - X\|_F^2 \; = \; \sum_{i > r} \sigma_i^2(M).$$

*Then, for every $r \geq 1$,*

$$\mathcal{E}_r^2(M) \; \leq \; \sum_{j=1}^{K} \|A_j\|_F^2 \; - \; \max_{1 \leq j \leq K} \sum_{i \leq r} \sigma_i^2(A_j). \tag{4}$$

*In particular, if $r \geq \max_{1 \leq j \leq K} \mathrm{rank}(A_j)$, then*

$$\mathcal{E}_r^2(M) \; \leq \; \sum_{j=1}^{K} \|A_j\|_F^2 \; - \; \max_{1 \leq j \leq K} \|A_j\|_F^2.$$

**Remark 1** (Single-anchor nature of the Weyl-based bound). *The Weyl-based upper bound in Theorem 1 relies on a* single anchor block *through the term*

$$\max_j \sum_{i \leq r} \sigma_i^2(A_j),$$

*and therefore cannot accumulate shared low-rank structure across multiple blocks. As a consequence, when compression arises from collective subspace alignment rather than dominance of a single matrix, the bound*

*may substantially overestimate the true rank-r approximation error. This explains the conservative behavior of the max-norm clustering algorithm observed in Section 4. A concrete example illustrating worst-case looseness is given in Appendix D.1.*

**Interpretation for clustering.** For any subset of indices $C \subseteq \{1, \ldots, K\}$, let

$$M_C := [A_j]_{j \in C}$$

denote the concatenation of matrices in that cluster. Applying Theorem 1 to $M_C$ yields

$$\mathcal{E}_r^2(M_C) \ \leq \ \sum_{j \in C} \|A_j\|_F^2 \ - \ \max_{j \in C} \sum_{i=1}^r \sigma_i^2(A_j).$$

In the common regime where $r$ is at least the rank of each block in the cluster (or large enough to recover each block essentially exactly when compressed alone), this simplifies to

$$\mathcal{E}_r^2(M_C) \ \leq \ \sum_{j \in C} \|A_j\|_F^2 \ - \ \max_{j \in C} \|A_j\|_F^2 \ = \ \sum_{j \in C \setminus \{j^\star\}} \|A_j\|_F^2,$$

where $j^\star$ is any index achieving the maximum Frobenius norm in the cluster. Thus, under this condition, the compression error of the concatenated cluster is bounded by the *sum of squared Frobenius norms of all blocks except the dominant one*. Intuitively, one large "anchor" matrix can absorb several smaller matrices at negligible additional error, as long as their total energy remains small.

This observation suggests a simple greedy strategy: for a given error tolerance $\varepsilon > 0$, we may form clusters by (i) selecting a high-energy block as an anchor, and (ii) attaching the lowest-energy remaining matrices as long as the sum of their squared norms does not exceed $\varepsilon^2$. For the details about the algorithm see Appendix E.1.

## 3.2 Residual-Based Global Bounds and Clustering

The Weyl-based bound in Section 3.1 depends only on individual blocks and is extremely fast to compute, but can be loose when the target rank $r$ exceeds the rank of each block. We now derive a sharper global bound based on the singular values of *incremental residuals*. This bound captures how each block contributes new directions beyond the span of all previously seen blocks. See the proof in Appendix C.2.

**Theorem 2** (Global incremental lower bound on singular values)**.** *Let*

$$M_K = \begin{bmatrix} A_1 & A_2 & \cdots & A_K \end{bmatrix} \in \mathbb{R}^{m \times (n_1 + \cdots + n_K)}$$

*be constructed incrementally from block matrices $A_i \in \mathbb{R}^{m \times n_i}$. Define $M_0 := 0$ and let $Q_{i-1}$ be any matrix with orthonormal columns spanning* range$(M_{i-1})$*. For each block $A_i$, define its orthogonal residual*

$$R_i := (I - Q_{i-1}Q_{i-1}^\top)A_i.$$

*Let $\widehat{R} \in \mathbb{R}^{m \times (n_1 + \cdots + n_K)}$ be the block concatenation of all residuals,*

$$\widehat{R} := \begin{bmatrix} R_1 & R_2 & \cdots & R_K \end{bmatrix},$$

*and let $\mu_1 \geq \mu_2 \geq \cdots \geq 0$ denote the singular values of $\widehat{R}$ (in non-increasing order, extended by zeros when necessary).*

*Then, for every $j \geq 1$,*

$$\sigma_j(M_K) \ \geq \ \mu_j.$$

*In particular, as new blocks are added, the sequence of lower bounds $\{\mu_j\}_{j \geq 1}$ is monotone non-decreasing and incorporates the contributions of all residual components discovered during the incremental construction.*

This theorem says that the singular values of the concatenated matrix $M_K$ are bounded from below by the singular values of the concatenated residuals $\widehat{R}$. Intuitively, every time a block contributes a component outside the span of all previously seen blocks, this "new direction" permanently lifts the spectrum of $M_K$. See the proof of the following corollary in Appendix C.3.

**Corollary 1** (Global upper bound on optimal SVD compression error). *In the setting of Theorem 2, let $M_K$ and $\mu_j$ be as above. For any target rank $r \geq 0$, define the optimal rank-$r$ SVD compression error of $M_K$ in the Frobenius norm by*

$$\mathcal{E}_r(M_K) := \min_{\mathrm{rank}(X) \leq r} \|M_K - X\|_F = \left( \sum_{j>r} \sigma_j(M_K)^2 \right)^{1/2},$$

*where $\sigma_1(M_K) \geq \sigma_2(M_K) \geq \cdots \geq 0$ are the singular values of $M_K$.*

*Then, for every $r \geq 0$,*

$$\mathcal{E}_r^2(M_K) \leq \sum_{i=1}^{K} \|A_i\|_F^2 - \sum_{j=1}^{r} \mu_j^2. \tag{5}$$

*In particular, as new blocks $A_i$ are added and $\widehat{R}$ accumulates more residual components, the quantity*

$$\sum_{i=1}^{K} \|A_i\|_F^2 - \sum_{j=1}^{r} \mu_j^2$$

*provides a global, monotonically decreasing upper bound on the best achievable rank-$r$ approximation error for $M_K$, computable from the per-block Frobenius norms and the singular values (or estimates) of $\widehat{R}$.*

To clarify when the residual-based bound of the Corollary 1 is exact, informative, or potentially conservative, we characterize its behavior under different structural regimes of the concatenated blocks in Appendix D.2.

**Remark 2** (Near-tightness under weak subspace overlap). *The residual-based bound of Corollary 1 becomes informative whenever each appended block contributes a non-negligible component outside the span of previously concatenated matrices. In such regimes, the leading singular values of the residual concatenation $\widehat{R}$ closely track those of the full matrix $M_K$, and the resulting upper bound on the truncated SVD error is empirically tight.*

*Conversely, when newly appended blocks lie largely within an already spanned subspace, residual energies are small and the bound may become conservative. This behavior reflects a fundamental limitation of worst-case guarantees based solely on subspace innovation and is intrinsic to concatenation-aware compression.*

**Interpretation for clustering.** For a cluster $C \subseteq \{1, \ldots, K\}$, let $M_C := [A_j]_{j \in C}$ denote its concatenated matrix. The residual-based construction processes the blocks in $C$ sequentially: when a new block $A_j$ is added, we remove the component already explained by the current cluster subspace and retain only the *residual $R_j$*, which captures directions not yet represented. The singular values $\mu_1(M_C) \geq \cdots \geq \mu_r(M_C)$ of the concatenated residual matrix $\widehat{R}_C$ therefore quantify the total energy of *new independent directions* introduced as the cluster grows.

Corollary 1 implies the bound

$$\mathcal{E}_r^2(M_C) \leq \sum_{j \in C} \|A_j\|_F^2 - \sum_{i=1}^{r} \mu_i^2(M_C),$$

which serves as a *merge certificate* for clustering: a block can be added to a cluster if the resulting upper bound on the rank-$r$ approximation error remains below a prescribed tolerance. In this way, clustering is driven by whether newly added blocks introduce too much unexplained energy beyond the current low-rank subspace.

Compared to the Weyl-based bound (Section 3.1), which depends primarily on the largest block, the residual-based bound aggregates the *innovation* contributed by all blocks. In particular, when multiple blocks contain components in different (approximately orthogonal) directions, the residual formulation captures their cumulative effect, leading to substantially tighter estimates of the achievable low-rank approximation error. For the algorithmic details, see Appendix E.2.

### 3.3 Approximate Compression Bound via Incremental Truncated SVD

The residual-based bound of Section 3.2 yields a provable upper bound on the optimal rank-$r$ compression error, but it only adds singular values of new directions without updating the singular values of the old ones. To obtain a more tight approximation, we maintain an orthonormal basis $Q_t$ for the approximate dominant left singular subspace of $M_t = [A_1, \ldots, A_t]$, together with a small Gram matrix $S_t$ whose eigenvalues track the dominant singular values. See Appendix B for more details.

Using these approximate singular values, we define a plug-in estimator of the SVD compression error. See the proof of the following corollary in Appendix C.4.

**Corollary 2** (Plug-in estimator of SVD compression error from incremental singular values). *Let*

$$M_K = \begin{bmatrix} A_1 & A_2 & \cdots & A_K \end{bmatrix} \in \mathbb{R}^{m \times (n_1 + \cdots + n_K)},$$

*and define the total Frobenius energy of $M_K$ by*

$$\|M_K\|_F^2 = \sum_{i=1}^{K} \|A_i\|_F^2.$$

*Let $\widetilde{\sigma}_1(M_K) \geq \widetilde{\sigma}_2(M_K) \geq \cdots \geq 0$ denote the approximate singular values produced by the truncated incremental scheme, i.e. the square roots of the retained eigenvalues of the Gram matrix in the compressed basis.*

*For any target rank $r \geq 0$, define the optimal rank-r SVD compression error*

$$\mathcal{E}_r(M_K) := \min_{\mathrm{rank}(X) \leq r} \|M_K - X\|_F = \left( \sum_{j > r} \sigma_j^2(M_K) \right)^{1/2},$$

*where $\sigma_1(M_K) \geq \sigma_2(M_K) \geq \cdots \geq 0$ are the true singular values of $M_K$. Then the quantity*

$$\widetilde{\mathcal{E}}_r(M_K) := \left( \sum_{i=1}^{K} \|A_i\|_F^2 - \sum_{j=1}^{r} \widetilde{\sigma}_j^2(M_K) \right)^{1/2}$$

*is a natural plug-in estimator of $\mathcal{E}_r(M_K)$. In particular, if the incremental scheme is executed without any truncation (so that $\widetilde{\sigma}_j(M_K) = \sigma_j(M_K)$ for all $j$), then*

$$\widetilde{\mathcal{E}}_r(M_K) = \mathcal{E}_r(M_K).$$

To provide intuition for the behavior of the plug-in estimator in Corollary 2, we first clarify how it is computed and used within the clustering procedure. The estimator is obtained via a truncated incremental scheme that processes the blocks of a cluster sequentially, maintaining a rank-$r$ approximation that is updated and truncated after each new block is appended. In the clustering algorithm, this estimate is used as a *merge criterion*: a block is added to a cluster if the predicted rank-$r$ approximation error remains below a prescribed tolerance.

**Remark 3** (Order-dependence and lack of guarantees). *The plug-in estimator depends on the order in which blocks are appended to a cluster, since the truncated incremental procedure maintains only a rank-r approximation of the evolving subspace and discards lower-energy directions after each update. When*

*the dominant subspace evolves smoothly and newly appended blocks contribute only small or well-aligned components, the estimator is typically accurate and empirically close to the true rank-r error.*

*However, the estimator does not provide a guaranteed bound. In particular, directions that are weak when first observed may be removed by truncation and later reappear with significant energy as additional blocks are added. In such cases, the estimator may either overestimate or underestimate the true error, depending on how the subspace evolves. This behavior is an intrinsic limitation of incremental low-rank approximation under fixed-rank constraints.*

**Interpretation and limitations.** The plug-in estimator can be interpreted as tracking how much energy is captured by a rank-$r$ subspace that is updated incrementally as blocks are added. It provides a fast, matrix-free surrogate for the true truncation error, enabling scalable clustering without forming the full concatenated matrix.

The estimator is most reliable when the dominant subspace of the cluster is stable, i.e., when new blocks lie largely within the existing span or introduce only low-energy orthogonal components. In this regime, the incremental subspace remains well aligned with the true top-$r$ singular subspace, and the estimated error closely matches the true approximation error.

The main limitation arises when truncation removes directions that later become important. If a direction is discarded early because it carries little energy, but is subsequently reinforced by later blocks, the incremental subspace may fail to recover it. This can lead to *underestimation* of the true error, resulting in negative slack $\widetilde{\mathcal{E}}_r(M_C) - \mathcal{E}_r(M_C) < 0$. A concrete construction demonstrating this behavior is provided in Appendix D.3.

Overall, the plug-in estimator trades theoretical guarantees for substantial computational efficiency, and should be viewed as a practical heuristic that complements the provable max-norm and residual-based clustering strategies. For algorithmic details, see Appendix E.3.

## 4 Evaluation

| Dataset | # Blocks | Original Shape | Final Shape | Size (GB) |
|---|---|---|---|---|
| Qualcomm MIMO SCM | 2468 | $(2, 20, 4, 50, 32)$ | $(12800, 20)$ | 2.35 |
| BigEarthNet-S1 | 10000 | $(120, 120, 2)$ | $(1440, 20)$ | 1.07 |
| PDEBench (Advection) | 5000 | $(1024, 201)$ | $(3072, 67)$ | 3.83 |
| SmolVLM2 256M | 333760 | – | $(768, 1)$ | 0.95 |

Table 1: Datasets used for evaluation (actual matrix shapes used in experiments).

We evaluate the proposed clustering–compression framework on four datasets with fundamentally different generative structure and spectral properties: (i) massive MIMO channel tensors with high-dimensional complex-valued geometry (Qualcomm AI Research, 2025), (ii) Sentinel-1 SAR satellite imagery (Clasen et al., 2025), (iii) PDE-generated advective flows (Takamoto et al., 2022), and (iv) SmolVLM2 256M model weights (Marafioti et al., 2025). These domains exhibit markedly different spectrum decay rates and inter-block alignment, allowing us to assess the robustness of the proposed algorithms beyond a single application regime. All datasets are reshaped into collections of fixed-shape matrices prior to clustering. For BigEarthNet-S1 and PDEBench, we randomly sample blocks from the full datasets to obtain representative subsets of manageable size. For the Qualcomm MIMO dataset, we use a single batch of channel realizations provided by the official release. For SmolVLM2, whose weight tensors have heterogeneous shapes, we reshape each weight tensor into a fixed-length vector and treat each vector as an individual block, excluding bias parameters. The exact matrix shapes, number of blocks, and dataset statistics used in the experiments are reported in Table 1.

We evaluate three clustering algorithms: (i) **max-norm clustering**, derived from the Weyl-type upper bound (Theorem 1, see Appendix E.1); (ii) **residual-based clustering**, evaluated with two sorting strategies: norm-descending and residual-ascending, based on Theorem 2 (see Appendix E.2); (iii) **approximate**

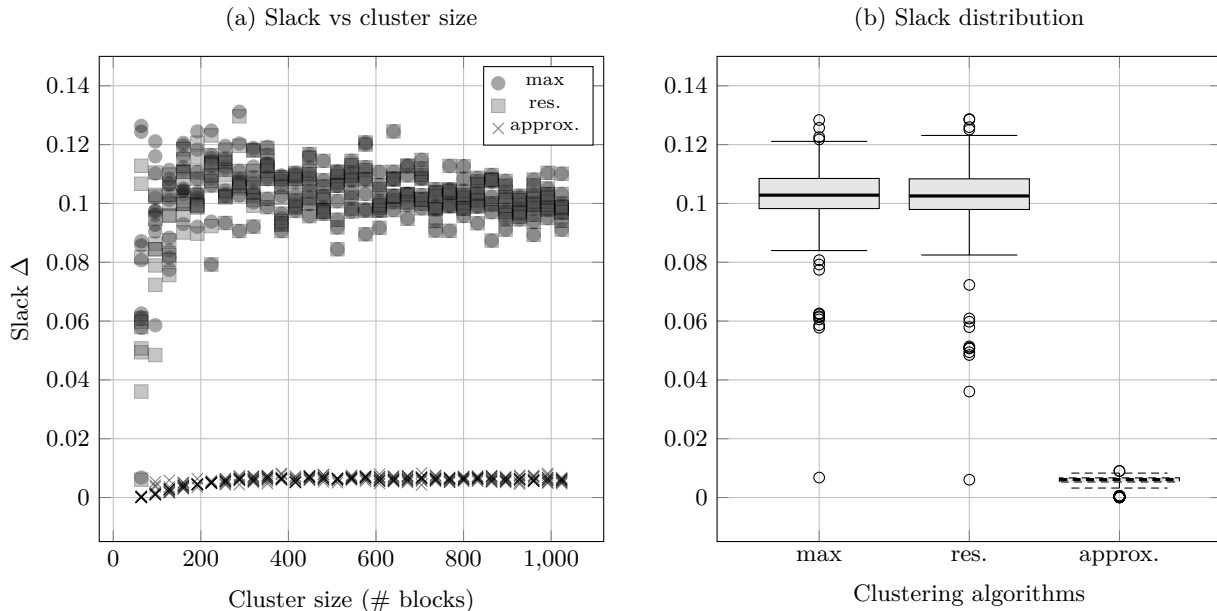

Figure 1: SmolVLM diagnostic of estimator conservativeness. Slack $\Delta = \widetilde{\mathcal{E}}_r - \mathcal{E}_r$ between predicted and true rank-$r$ SVD reconstruction error. For each cluster size and estimator, all 10 independent trials (uniform block samples) are shown. (a) slack versus cluster size; (b) empirical slack distribution across estimators.

**incremental clustering**, also evaluated with both sorting strategies, derived from Corollary 2 (see Appendix E.3). All algorithms depend on a target relative reconstruction error and a target approximation rank.

All experiments were run under identical hardware conditions: Intel Core i5-13600KF (20 threads) with 64 GB RAM.

**Reconstruction error.** Let $X \in \mathbb{R}^{M \times N}$ denote the concatenated matrix formed from all blocks assigned to a cluster, and let $\hat{X}$ be its low-rank reconstruction after clustering and decoding. We measure reconstruction error using the relative Frobenius norm error

$$\varepsilon_{\text{rel}} = \frac{\|X - \hat{X}\|_F}{\|X\|_F}. \tag{6}$$

For the max-norm and residual-based clustering algorithms, Theorems 1 and 2 guarantee that the relative reconstruction error does not exceed the user-specified error constraint. The approximate incremental algorithm does not provide a formal guarantee, however, no violations of the prescribed error thresholds were observed in any experiment. This empirical stability is consistent with the presence of strong spectral gaps in the tested datasets.

**Compression ratio.** Assume that each block has shape $(m, n)$ and that a cluster contains $K$ blocks approximated with target rank $r$. The uncompressed representation stores $Kmn$ parameters. After compression, a shared basis requires $mr$ parameters, while per-block coefficients require $Knr$ parameters, resulting in a total of $r(m + Kn)$ parameters. The compression ratio is defined as the ratio between the number of parameters before and after compression.

**Estimator conservativeness diagnostic.** In addition to end-to-end compression performance, we explicitly evaluate how conservative the proposed error estimators are relative to the true truncated SVD error. For a fixed target rank $r$, we uniformly sample subsets of blocks of increasing cluster size, form their concatenation, and compute the true optimal rank-$r$ reconstruction error $\mathcal{E}_r$ via an explicit SVD. For each sampled

| Method | Qualcomm | BigEarthNet | PDEBench | SmolVLM2 | Wall time$^\dagger$ |
|---|---|---|---|---|---|
| max norm | 2.138 | 1.000$^*$ | 1.004$^*$ | 1.581 | 1× |
| res. (norm sorting) | 2.276 | 1.000$^*$ | 1.005$^*$ | 1.588 | 2168× |
| res. (res. sorting) | 2.243 | 1.000$^*$ | 1.002$^*$ | 1.547 | 4620× |
| approx. (norm sorting) | 2.301 | 1.789 | **45.434** | 1.642 | **100×** |
| approx. (res. sorting) | **2.334** | **1.870** | **45.434** | **1.807** | 1792× |

$^*$ Clustering failed; no compression achieved (compression ratio $\approx 1.0$).
$^\dagger$ Wall time is reported relative to a baseline of 130 ms.

Table 2: Compression ratios across datasets and worst-case clustering wall time. Wall-clock time for clustering (excluding decomposition) is reported relative to the max-norm method (higher is slower). Clustering were run with 5% relative reconstruction error constraint for Qualcomm, BigEarthNet and PDEBench and with 20% for SmolVLM. And target rank is fixed to 20 for Qualcomm and BigEarthNet, to 67 for PDEBench and to 32 for SmolVLM.

subset, we also compute the corresponding predicted error $\tilde{\mathcal{E}}_r$ produced by the max-norm, residual-based, and approximate incremental estimators. For each cluster size and estimator, this procedure is repeated over 10 independent random trials with different block samples; *all individual trial outcomes are shown* in Figure 1. We report the resulting *slack* $\Delta = \tilde{\mathcal{E}}_r - \mathcal{E}_r$, which directly measures estimator conservativeness.

As predicted by Theorems 1-2, the max-norm and residual-based estimators remain strictly conservative across all trials, exhibiting a consistently positive slack that is largely insensitive to cluster size.

In contrast, the approximate incremental estimator produces substantially smaller slack (often close to zero), indicating a much tighter but non-guaranteed estimate of the true error. We note that, although underestimation of the true error is possible in principle (see the formal construction in Appendix D.3), we did not observe negative slack in these experiments, even under random sampling of blocks. A more detailed empirical analysis of slack distributions across datasets is provided in Appendix F.

The observed dispersion across trials reflects variability induced by block selection while preserving a clear qualitative separation between estimators.

**Clustering results.** Table 2 reports compression ratios and worst-case wall-clock times for the clustering stage only, excluding the cost of the final low-rank decompositions, across all datasets. Entries marked with $*$ indicate failure to achieve compression, defined as the inability to form clusters larger than individual blocks, resulting in compression ratios close to one. This behavior is not caused by numerical instability, but arises when conservative error bounds prevent aggregation of blocks into shared low-rank subspaces.

The results reveal a clear trade-off between computational efficiency, compression performance, and theoretical guarantees. Max-norm clustering is consistently the fastest method, but yields conservative clustering due to loose worst-case bounds, limiting achievable compression. Residual-based clustering enforces strict reconstruction guarantees and produces stable results when feasible, but incurs several orders of magnitude higher runtime. Approximate incremental clustering achieves the highest compression, while remaining substantially faster than exact residual-based methods.

**Downstream implications.** While our primary focus is on reconstruction error, we additionally evaluate how compression affects a simple downstream forecasting task on PDEBench. The results (see Appendix G) show that low reconstruction error correlates with strong downstream performance up to a sharp degradation threshold, and that naive full concatenation can lead to catastrophic performance loss despite high compression.

**Hyperparameter sensitivity.** We analyze the dependence of compression performance on the target reconstruction error and target rank. Figure 2 shows results for the Qualcomm MIMO dataset using approximate clustering with residual-based sorting, and reports results for SmolVLM2 weights using approximate clustering with norm-based sorting. Across both datasets, compression increases approximately linearly with

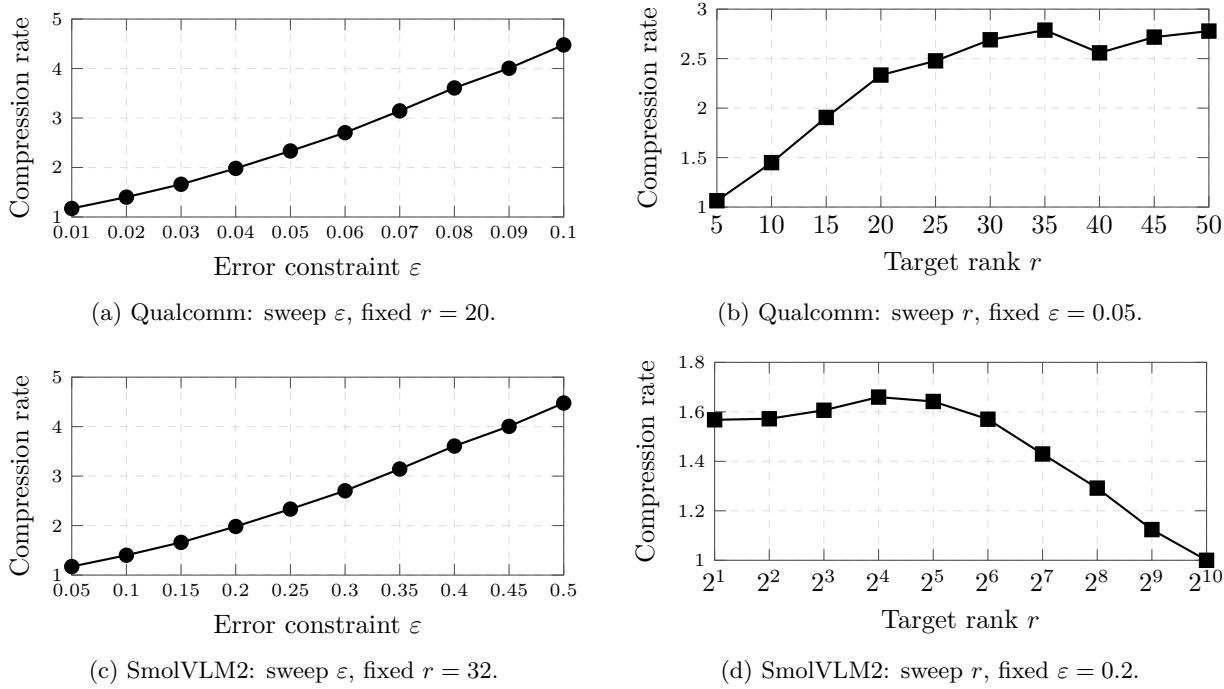

Figure 2: Compression rates under feasibility constraints. (a) Qualcomm dataset: fixed target rank and varying error constraint. (b) Qualcomm dataset: fixed error constraint and varying target rank. (c) SmolVLM2 model weights: fixed target rank and varying error constraint. (d) SmolVLM2 model weights: fixed error constraint and varying target rank.

the allowed reconstruction error, consistent with low-rank approximation theory. In contrast, the dependence on target rank is non-monotonic: increasing rank does not necessarily improve compression. This behavior reflects the interaction between rank selection and inter-block subspace alignment, highlighting that rank is not merely a capacity parameter but must be chosen carefully to balance expressivity and shared structure.

**Rank selection.** The target rank $r$ plays a dual role in our framework: for a fixed cluster it controls truncation error, while simultaneously influencing the clustering outcome by determining which merges satisfy the error constraint. As a result, the overall compression ratio is generally *non-monotonic* in $r$, since varying $r$ changes the feasible set of cluster assignments rather than only refining a fixed approximation. Crucially, our approach does not require access to the singular values of the concatenated matrices. Instead, rank selection relies on the same surrogate quantities used for clustering decisions. In practice, we treat $r$ as a hyperparameter and evaluate a small set of candidate values, selecting the configuration that achieves the best compression under the prescribed error constraint.

**Classical clustering baselines.** We additionally compare against baseline clustering methods on the Qualcomm dataset. Table 3 reports results for random clustering and k-means with varying numbers of clusters and for HDBSCAN. These methods are fundamentally misaligned with the concatenated SVD compression objective. Classical clustering algorithms optimize geometric distortion in the ambient space and provide no mechanism to control spectral reconstruction error after low-rank decoding. As a result, high compression is achieved only at the cost of unacceptable and highly unstable reconstruction error, while density-based clustering fails entirely by labeling all points as outliers. This confirms that standard clustering techniques are unsuitable for controlled low-rank compression of concatenated matrices.

| Method | Compression | Rel. error | Outcome |
|---|---|---|---|
| Random clustering ($K$=10) | 178.1 | $0.704 \pm 0.034$ | High compression and error |
| Random clustering ($K$=100) | 23.8 | $0.363 \pm 0.110$ | Moderate compression, high error |
| Random clustering ($K$=1000) | 2.46 | $0.199 \pm 0.134$ | Low compression, moderate error |
| k-means ($K$=10) | 178.3 | 0.886 | High compression and error |
| k-means ($K$=100) | 23.8 | $0.322 \pm 0.343$ | High unstable error |
| k-means ($K$=1000) | 2.46 | $0.881 \pm 0.098$ | High error |
| HDBSCAN (min cluster size = 2) | – | – | All points labeled as outliers |
| **Ours (approx. residual)** | 2.334 | $\mathbf{0.044 \pm 0.006}$ | Low stable error |

Table 3: Classical clustering baselines and proposed approximate method on the Qualcomm dataset.

## 5 Conclusion

We presented a theory-driven framework for compression-aware clustering of matrix collections under explicit SVD reconstruction constraints. By analyzing the spectral behavior of horizontally concatenated matrices, we derived global upper bounds on truncated SVD reconstruction error from lower bounds on singular value growth. These results enable principled decisions about which matrices can be safely grouped and compressed together.

Building on this theory, we proposed three clustering algorithms that span a spectrum of trade-offs between computational efficiency, tightness of guarantees, and scalability. The max-norm method provides a fast but conservative baseline, while the residual-based method yields a substantially tighter and fully deterministic certificate by explicitly tracking subspace innovation across blocks. Although computationally more expensive, the residual-based approach serves as a reference method for guaranteed compression and enables merges that cannot be justified by worst-case bounds alone. In contrast, the approximate incremental estimator achieves significantly higher compression and efficiency in practice, but does not provide formal guarantees: as we demonstrate, it may in principle underestimate the true error due to truncation effects, even though such behavior was not observed empirically in our experiments. Unlike existing heuristic or distance-based approaches, the proposed methods directly tie cluster formation to low-rank approximation error, providing explicit and interpretable accuracy control.

Together, these contributions establish concatenation-aware SVD compression as a unifying perspective on matrix clustering and low-rank approximation, clarifying the interplay between guarantees and practical performance, and providing both theoretical foundations and scalable algorithms for compressing large collections of matrices in modern machine learning and scientific computing pipelines.

**Discussion and future work.** Throughout this work, the target rank is treated as a fixed hyperparameter. In practice, the optimal choice of $r$ depends on the interaction between truncation error and cluster formation, and may vary across clusters. Importantly, the overall compression is generally non-monotonic in $r$, since changing $r$ alters not only the approximation within each cluster but also which merges satisfy the error constraint. A principled approach to selecting $r$ without solving the full clustering problem remains an open question. In particular, joint optimization of rank selection and clustering, with $r$ adapting during merging, is a promising direction for future work.

The present framework operates in an offline setting where all matrices are available in advance. In many applications, however, new data arrive sequentially. Extending compression-aware clustering to an incremental setting, where each new matrix must be assigned to an existing cluster or used to initialize a new cluster based on predicted SVD compression error, is a natural and practically important direction for future work. Such an extension would require efficient per-cluster error estimation and dynamic cluster management under streaming updates.

The current framework measures reconstruction quality using the Frobenius norm. This choice is motivated by the compression objective: for truncated SVD, the Frobenius error corresponds to the total discarded spectral energy, which admits an additive interpretation across singular directions and aligns naturally with memory–distortion trade-offs. In contrast, spectral norm controls only the largest residual singular value and therefore captures worst-case directional error rather than aggregate reconstruction quality. As a result, two clusterings may exhibit similar spectral error while differing substantially in total reconstruction fidelity. Moreover, the proposed clustering criteria rely on energy-based certificates that accumulate contributions across multiple directions, which are inherently tied to Frobenius-type quantities. Extending the framework to alternative norms, such as spectral norm, would require different merge certificates that directly control the $(r + 1)$-th singular value of concatenated matrices, and is left for future work.

Beyond pure reconstruction, low-rank representations are often used as intermediate features for downstream tasks. In such settings, reconstruction error may not fully capture task-relevant information. Incorporating task-aware objectives into compression-driven clustering while retaining spectral guarantees remains an open and challenging problem.

Finally, we note that the approximate incremental estimator is closely related to randomized sketching and streaming low-rank approximation methods. In principle, it could be replaced by a randomized SVD or range-finder approach, yielding $(1 + \varepsilon)$-approximate rank-$r$ reconstructions in Frobenius norm with high probability and near-linear computational cost. In the clustering setting, this would lead to *probabilistic* merge criteria and end-to-end guarantees that hold with high probability, in contrast to the deterministic worst-case guarantees provided by the residual-based method. However, randomized approximations do not naturally yield conservative upper bounds on the residual energy, which are required to ensure safe merges under a prescribed error tolerance. Designing sketch-based clustering procedures that preserve certificate-based guarantees while improving computational efficiency is an interesting direction for future work.

### Acknowledgments

The authors confirm that there is no conflict of interest and acknowledges financial support by the Simons Foundation grant (SFI-PD-Ukraine-00014586, M.S.) and the project 0125U000299 of the National Academy of Sciences of Ukraine. We also express our gratitude to the Armed Forces of Ukraine for their protection, which has made this research possible.

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

# A  Preliminaries

This section summarizes the mathematical tools used throughout the paper. Our theoretical results rely on two classical components of matrix analysis: (i) perturbation inequalities for eigenvalues and singular values, and (ii) optimality properties of truncated SVD. We also introduce the notation and identities underlying our incremental representation of concatenated matrices.

## A.1  Spectral Monotonicity and Perturbation Bounds

A key component in our analysis is the fact that appending a matrix block to an existing concatenation can only *increase* its singular values. This follows from a classical form of Weyl's monotonicity theorem for Hermitian matrices. Letting $M = [A_1, \ldots, A_t]$ and $M' = [A_1, \ldots, A_t, A_{t+1}]$, we have $M'M'^\top = MM^\top + A_{t+1}A_{t+1}^\top$, so the update is positive semidefinite.

**Theorem 3** (Weyl Monotonicity; cf. Cor. 4.9 in Stewart & Sun (1990)). *Let $A, E \in \mathbb{R}^{n \times n}$ be Hermitian and write their eigenvalues in nonincreasing order,*

$$\lambda_1(A) \geq \cdots \geq \lambda_n(A), \qquad \lambda_1(E) \geq \cdots \geq \lambda_n(E).$$

*Let $\tilde{\lambda}_1 \geq \cdots \geq \tilde{\lambda}_n$ be the eigenvalues of $A + E$. Then, for all $i = 1, \ldots, n$,*

$$\tilde{\lambda}_i \in \big[\lambda_i(A) + \lambda_n(E), \, \lambda_i(A) + \lambda_1(E)\big].$$

*In particular, if $E \succeq 0$, then*

$$\tilde{\lambda}_i \geq \lambda_i(A), \qquad i = 1, \ldots, n.$$

Since the singular values of a matrix $M$ are the square roots of the eigenvalues of $MM^\top$, the theorem immediately yields monotonicity of singular values under horizontal concatenation.

## A.2  Optimality of Truncated SVD

We frequently use the classical Eckart–Young–Mirsky theorem, which states that the best rank-$r$ approximation of a matrix in Frobenius norm is obtained by truncating its singular value decomposition.

**Theorem 4** (Eckart–Young–Mirsky (Eckart & Young, 1936; Mirsky, 1960)). *Let $A \in \mathbb{R}^{m \times n}$ have singular values $\sigma_1(A) \geq \sigma_2(A) \geq \cdots \geq 0$, and let*

$$A = U\Sigma V^\top, \qquad \Sigma = \mathrm{diag}(\sigma_1, \ldots, \sigma_p), \quad p = \min\{m, n\}.$$

*For $r \in \{0, \ldots, p\}$, define the rank-r truncated SVD*

$$A_r := U \begin{bmatrix} \mathrm{diag}(\sigma_1, \ldots, \sigma_r) & 0 \\ 0 & 0 \end{bmatrix} V^\top.$$

*Then $A_r$ is the best rank-r approximation to $A$ in the Frobenius norm:*

$$\|A - A_r\|_F = \min_{\mathrm{rank}(X) \leq r} \|A - X\|_F = \left(\sum_{j > r} \sigma_j(A)^2\right)^{1/2}.$$

This theorem underlies all of our compression error formulas.

### A.3 Incremental Representation of Concatenated Matrices

Let $M_t = [A_1, \ldots, A_t]$ denote the concatenation of $t$ blocks. When a new block $A_{t+1}$ is appended, the Gram matrix updates as

$$M_{t+1}M_{t+1}^\top = M_t M_t^\top + A_{t+1}A_{t+1}^\top,$$

which is a rank-$\leq n_{t+1}$ positive semidefinite perturbation. Instead of storing the full $m \times m$ matrix $M_t M_t^\top$, we maintain an orthonormal basis $Q_t$ for the current column space of $M_t$ and pose $M_t M_t^\top$ by this basis via a small Gram matrix $S_t$, i.e.,

$$M_t M_t^\top = Q_t S_t Q_t^\top.$$

When $A_{t+1}$ arrives, we decompose it by $Q_t$ and the residual orthogonal to $Q_t$, as follows

$$A_{t+1} = Q_t Y_{t+1} + R_{t+1}, \qquad R_{t+1} = (I - Q_t Q_t^\top)A_{t+1}.$$

Expanding the basis by the columns of $R_{t+1}$ yields an updated orthonormal matrix $Q_{t+1}$, and the new Gram matrix $S_{t+1}$ takes a block form constructed from $S_t$, $Y_{t+1}$, and the SVD of $R_{t+1}$. This identity, stated formally in Lemma 1 and proved in Appendix B, is exact and well known in the incremental PCA/SVD literature. It provides the algebraic foundation for our blockwise concatenation analysis.

After expanding the basis and updating $S_{t+1}$, its size grows by the rank of $R_{t+1}$. To control complexity, we compress back to the rank $r$ by retaining only the top $r$ eigenpairs of $S_{t+1}$. By Theorem 4, this produces the optimal rank-$r$ approximation *within the expanded subspace*. The approximate singular values of $M_{t+1}$ are then given by the square roots of the retained eigenvalues.

The only source of approximation in our incremental estimator is this truncation step, all other steps are exact. A full characterization of the truncation and its implications is provided in Corollary 3.

**Stability of the incremental estimator.** Although the truncated incremental scheme does not provide deterministic upper or lower bounds on the true truncated SVD error, its approximation quality is governed by classical subspace stability principles. In particular, when the singular value gap $\sigma_r(M_t) - \sigma_{r+1}(M_t)$ is sufficiently large, truncation preserves the dominant invariant subspace up to small perturbations, and the retained eigenvalues of the compressed Gram matrix remain close to the true leading singular values. This behavior is well understood in incremental and randomized SVD literature, where approximation error scales with the truncation gap and the energy of discarded components.

## B  Incremental SVD identities

This appendix contains the algebraic identities used in Section 3.3 to derive our incremental estimator for the top singular values of a concatenated matrix. These results are classical in incremental PCA/SVD, but we provide them for completeness and to make the paper self-contained.

**Lemma 1** (Exact incremental Gram factorisation for concatenated blocks)**.** *Let $M_t = [A_1, \ldots, A_t] \in \mathbb{R}^{m \times n_t}$ be the horizontal concatenation of the first $t$ blocks, and suppose that for some $r_t$ we have an exact factorisation*

$$M_t M_t^\top = Q_t S_t Q_t^\top,$$

*where $Q_t \in \mathbb{R}^{m \times r_t}$ has orthonormal columns and $S_t \in \mathbb{R}^{r_t \times r_t}$ is symmetric positive semidefinite. Let a new block $A_{t+1} \in \mathbb{R}^{m \times k}$ be given and define*

$$Y := Q_t^\top A_{t+1}, \qquad R := A_{t+1} - Q_t Y.$$

*Compute a thin QR decomposition of the residual,*

$$R = Q_{\mathrm{res}} B,$$

*with $Q_{\mathrm{res}} \in \mathbb{R}^{m \times r_{\mathrm{res}}}$ having orthonormal columns and $B \in \mathbb{R}^{r_{\mathrm{res}} \times k}$, and set*

$$Q_{t+1} := \begin{bmatrix} Q_t & Q_{\mathrm{res}} \end{bmatrix} \in \mathbb{R}^{m \times (r_t + r_{\mathrm{res}})}.$$

*Define the extended Gram matrix*

$$S_{t+1} := \begin{bmatrix} S_t + YY^\top & YB^\top \\ BY^\top & BB^\top \end{bmatrix} \in \mathbb{R}^{(r_t+r_{\mathrm{res}})\times(r_t+r_{\mathrm{res}})}.$$

*Then $Q_{t+1}$ has orthonormal columns and*

$$M_{t+1}M_{t+1}^\top = Q_{t+1}S_{t+1}Q_{t+1}^\top,$$

*where $M_{t+1} := [M_t, A_{t+1}]$.*

*Proof.* By construction, $Q_t$ has orthonormal columns and $Q_{\mathrm{res}}$ is the $Q$-factor of a thin QR decomposition of the residual $R$. Moreover, $R = (I - Q_tQ_t^\top)A_{t+1}$ lies in the orthogonal complement of range$(Q_t)$, hence $Q_t^\top Q_{\mathrm{res}} = 0$, and therefore $Q_{t+1}$ has orthonormal columns.

We first expand the new Gram matrix explicitly:

$$M_{t+1}M_{t+1}^\top = M_tM_t^\top + A_{t+1}A_{t+1}^\top.$$

Using the decomposition $A_{t+1} = Q_tY + R$ and $M_tM_t^\top = Q_tS_tQ_t^\top$, we get

$$\begin{aligned} M_{t+1}M_{t+1}^\top &= Q_tS_tQ_t^\top + (Q_tY + R)(Q_tY + R)^\top \\ &= Q_tS_tQ_t^\top + Q_tYY^\top Q_t^\top + Q_tYR^\top + RY^\top Q_t^\top + RR^\top. \end{aligned}$$

The orthogonality relation $Q_t^\top R = 0$ implies $R = Q_{\mathrm{res}}B$ for some $B$, namely the $R$-factor of the QR decomposition. Substituting this into the above yields

$$\begin{aligned} M_{t+1}M_{t+1}^\top &= Q_t(S_t + YY^\top)Q_t^\top + Q_tYB^\top Q_{\mathrm{res}}^\top + Q_{\mathrm{res}}BY^\top Q_t^\top + Q_{\mathrm{res}}BB^\top Q_{\mathrm{res}}^\top \\ &= \begin{bmatrix} Q_t & Q_{\mathrm{res}} \end{bmatrix} \begin{bmatrix} S_t + YY^\top & YB^\top \\ BY^\top & BB^\top \end{bmatrix} \begin{bmatrix} Q_t & Q_{\mathrm{res}} \end{bmatrix}^\top \\ &= Q_{t+1}S_{t+1}Q_{t+1}^\top, \end{aligned}$$

as claimed. $\qquad\square$

**Corollary 3** (Truncated incremental top-$r$ approximation). *In the setting of Lemma 1, let*

$$S_{t+1} = U\Lambda U^\top$$

*be an eigendecomposition of $S_{t+1}$ with eigenvalues ordered as $\lambda_1 \geq \lambda_2 \geq \cdots \geq \lambda_{r_t+r_{\mathrm{res}}} \geq 0$. For a target rank $r \leq r_t + r_{\mathrm{res}}$, define*

$$U_r := \begin{bmatrix} u_1 & \cdots & u_r \end{bmatrix}, \qquad \Lambda_r := \mathrm{diag}(\lambda_1,\ldots,\lambda_r).$$

*Set*

$$\widetilde{Q}_{t+1} := Q_{t+1}U_r \in \mathbb{R}^{m\times r}, \qquad \widetilde{S}_{t+1} := \Lambda_r \in \mathbb{R}^{r\times r}.$$

*Then:*

1. *The matrix $\widetilde{G}_{t+1} := \widetilde{Q}_{t+1}\widetilde{S}_{t+1}\widetilde{Q}_{t+1}^\top$ is the best rank-$r$ approximation to $G_{t+1} := M_{t+1}M_{t+1}^\top$ within the subspace* range$(Q_{t+1})$, *in both spectral and Frobenius norms, i.e.*

$$\|G_{t+1} - \widetilde{G}_{t+1}\|_F^2 = \sum_{j>r}\lambda_j(S_{t+1}), \qquad \|G_{t+1} - \widetilde{G}_{t+1}\|_2 = \lambda_{r+1}(S_{t+1}).$$

2. *The top $r$ approximate singular values of $M_{t+1}$ produced by this scheme are*

$$\widetilde{\sigma}_j(M_{t+1}) := \sqrt{\lambda_j(S_{t+1})}, \qquad j = 1,\ldots,r.$$

*Proof.* By Lemma 1, $G_{t+1} = Q_{t+1} S_{t+1} Q_{t+1}^\top$ with $Q_{t+1}$ orthonormal. Any rank-$r$ approximation $\widehat{G}$ whose range is contained in range($Q_{t+1}$) can be written as $\widehat{G} = Q_{t+1} X Q_{t+1}^\top$ with rank($X$) $\leq r$. Because $Q_{t+1}$ is orthonormal, the Frobenius and spectral norms satisfy

$$\|G_{t+1} - \widehat{G}\| = \|S_{t+1} - X\|,$$

for both $\|\cdot\|_F$ and $\|\cdot\|_2$. By the Eckart–Young–Mirsky theorem (Theorem 4), the best rank-$r$ approximation to $S_{t+1}$ in Frobenius and spectral norms is $X = U_r \Lambda_r U_r^\top$, with errors $\sum_{j>r} \lambda_j(S_{t+1})$ and $\lambda_{r+1}(S_{t+1})$, respectively. Substituting $X = U_r \Lambda_r U_r^\top$ and noting that $Q_{t+1} U_r = \widetilde{Q}_{t+1}$ and $\Lambda_r = \widetilde{S}_{t+1}$ gives the first claim.

The second statement is just the observation that the eigenvalues of $\widetilde{G}_{t+1}$ equal $\lambda_1(S_{t+1}), \ldots, \lambda_r(S_{t+1})$, so the corresponding approximate singular values of $M_{t+1}$ are their square roots. $\square$

## C  Proofs

### C.1  Proof of Theorem 1

*Proof.* Write the singular values of a matrix $X$ in the non-increasing order as $\sigma_1(X) \geq \cdots \geq \sigma_{\min(m,n)}(X)$. By the Eckart–Young–Mirsky theorem (Theorem 4),

$$\mathcal{E}_r^2(M) = \sum_{i>r} \sigma_i^2(M) = \|M\|_F^2 - \sum_{i=1}^r \sigma_i^2(M).$$

Since $M = [A_1, \ldots, A_K]$ is a horizontal concatenation, its Frobenius norm decomposes to

$$\|M\|_F^2 = \sum_{j=1}^K \|A_j\|_F^2.$$

Thus

$$\mathcal{E}_r^2(M) = \sum_{j=1}^K \|A_j\|_F^2 - \sum_{i=1}^r \sigma_i^2(M). \tag{7}$$

We now relate the singular values of $M$ to those of the blocks $A_j$. Define

$$B_j := A_j A_j^\top \succeq 0, \qquad j = 1, \ldots, K,$$

so that

$$MM^\top = \sum_{j=1}^K A_j A_j^\top = \sum_{j=1}^K B_j.$$

Fix $k \in \{1, \ldots, K\}$ and write

$$MM^\top = B_k + C_k, \qquad C_k := \sum_{\substack{j=1 \\ j \neq k}}^K B_j \succeq 0.$$

By the Weyl monotonicity (Theorem 3), if $H, G$ are Hermitian with $G \succeq 0$ and $\lambda_1(\cdot) \geq \cdots \geq \lambda_n(\cdot)$ denote eigenvalues in nonincreasing order, then

$$\lambda_i(H + G) \geq \lambda_i(H), \qquad i = 1, \ldots, n.$$

Applying this with $H = B_k$ and $G = C_k$ yields

$$\lambda_i(MM^\top) = \lambda_i(B_k + C_k) \geq \lambda_i(B_k), \qquad i = 1, \ldots, m.$$

Using $\sigma_i^2(X) = \lambda_i(XX^\top)$, we obtain

$$\sigma_i^2(M) = \lambda_i(MM^\top) \geq \lambda_i(B_k) = \sigma_i^2(A_k), \qquad i = 1, \ldots, \min(m, n_k).$$

Summing over $i = 1, \ldots, r$ gives

$$\sum_{i=1}^{r} \sigma_i^2(M) \geq \sum_{i=1}^{r} \sigma_i^2(A_k) \qquad \text{for every } k = 1, \ldots, K.$$

Hence

$$\sum_{i=1}^{r} \sigma_i^2(M) \geq \max_{1 \leq j \leq K} \sum_{i=1}^{r} \sigma_i^2(A_j). \tag{8}$$

Combining equation 7 and equation 8, we obtain

$$\mathcal{E}_r^2(M) = \sum_{j=1}^{K} \|A_j\|_F^2 - \sum_{i=1}^{r} \sigma_i^2(M) \leq \sum_{j=1}^{K} \|A_j\|_F^2 - \max_{1 \leq j \leq K} \sum_{i=1}^{r} \sigma_i^2(A_j),$$

which is exactly equation 4. If $r \geq \mathrm{rank}(A_j)$ for all $j$, then $\sum_{i=1}^{r} \sigma_i^2(A_j) = \|A_j\|_F^2$, which yields the stated special case. $\qquad\square$

### C.2 Proof of Theorem 2

*Proof.* We prove the result by comparing the Gram matrix of $M_K$ with the Gram matrix of the concatenated residuals $\widehat{R}$, and then invoking the eigenvalue monotonicity for Hermitian matrices.

**Step 1: Gram matrix of the final concatenation.** Define the (symmetric, positive semidefinite) Gram matrix

$$X := M_K M_K^\top = \begin{bmatrix} A_1 & A_2 & \cdots & A_K \end{bmatrix} \begin{bmatrix} A_1 & A_2 & \cdots & A_K \end{bmatrix}^\top = \sum_{i=1}^{K} A_i A_i^\top,$$

where the singular values of $M_K$ are related to the eigenvalues of $X$,

$$\sigma_j^2(M_K) = \lambda_j(X), \qquad j = 1, \ldots, m,$$

and the eigenvalues are ordered non-increasingly: $\lambda_1(X) \geq \lambda_2(X) \geq \cdots \geq \lambda_m(X) \geq 0$.

**Step 2: Decomposition of each block into "old span + residual".** By construction, $Q_{i-1}$ has orthonormal columns spanning $\mathrm{range}(M_{i-1})$. We can therefore write each block $A_i$ as a sum of a part in the span of $Q_{i-1}$ and an orthogonal residual. More precisely, define

$$B_i := Q_{i-1}^\top A_i \in \mathbb{R}^{r_{i-1} \times n_i},$$

where $r_{i-1} = \mathrm{rank}(M_{i-1}) = \mathrm{cols}(Q_{i-1})$, and recall that

$$R_i := (I - Q_{i-1}Q_{i-1}^\top)A_i.$$

Then we have the orthogonal decomposition

$$A_i = Q_{i-1}B_i + R_i,$$

with

$$\mathrm{range}(Q_{i-1}B_i) \subseteq \mathrm{range}(Q_{i-1}) = \mathrm{range}(M_{i-1}), \qquad \mathrm{range}(R_i) \subseteq \mathrm{range}(M_{i-1})^\perp.$$

In particular, $Q_{i-1}B_i$ and $R_i$ have orthogonal column spaces, therefore, their Gram matrices have no cross terms:

$$(Q_{i-1}B_i)(R_i)^\top = Q_{i-1}B_i R_i^\top = 0, \qquad R_i(Q_{i-1}B_i)^\top = 0.$$

Using the decomposition $A_i = Q_{i-1}B_i + R_i$, we expand

$$A_iA_i^\top = (Q_{i-1}B_i + R_i)(Q_{i-1}B_i + R_i)^\top = Q_{i-1}B_iB_i^\top Q_{i-1}^\top + R_iR_i^\top,$$

since the cross terms vanish by the orthogonality noted above. Therefore,

$$A_iA_i^\top - R_iR_i^\top = Q_{i-1}B_iB_i^\top Q_{i-1}^\top \succeq 0, \tag{9}$$

i.e., $A_iA_i^\top \succeq R_iR_i^\top$ in the Loewner (positive semidefinite) order.

Summing equation 9 over $i = 1, \ldots, K$ gives

$$\sum_{i=1}^{K} A_iA_i^\top \succeq \sum_{i=1}^{K} R_iR_i^\top.$$

By the definition,

$$X = M_K M_K^\top \succeq Y := \sum_{i=1}^{K} R_iR_i^\top. \tag{10}$$

**Step 3: Orthogonality of the residual ranges.** We now show that the column spaces of the residuals $R_i$ are mutually orthogonal. This is a consequence of the incremental construction.

Since each $R_j$ is a linear combination of the columns of $A_j$, we have $\text{range}(R_j) \subseteq \text{range}(A_j)$. Because $\text{range}(M_{i-1})$ contains $A_1, \ldots, A_{i-1}$, it follows that

$$\text{range}(R_j) \subseteq \text{range}(M_{i-1}), \qquad j < i.$$

On the other hand,

$$R_i = (I - Q_{i-1}Q_{i-1}^\top)A_i$$

lies in the orthogonal complement of the $\text{range}(Q_{i-1})$, which is equal to the orthogonal complement of $\text{range}(M_{i-1})$. Thus

$$\text{range}(R_i) \subseteq \text{range}(M_{i-1})^\perp \quad \Rightarrow \quad \text{range}(R_i) \perp \text{range}(R_j) \text{ for all } j < i.$$

By symmetry of the indices, this shows that the subspaces $\text{range}(R_1), \ldots, \text{range}(R_K)$ are pairwise orthogonal.

**Step 4: Gram matrix of the concatenated residuals.** Consider the concatenated residual matrix

$$\widehat{R} := \begin{bmatrix} R_1 & R_2 & \cdots & R_K \end{bmatrix} \quad \text{and} \quad \widehat{R}\widehat{R}^\top = \sum_{i=1}^{K} R_iR_i^\top = Y.$$

Because the column spaces of $R_i$ are pairwise orthogonal, there exists an orthonormal basis of $\mathbb{R}^m$ in which $Y$ becomes block diagonal with blocks $R_iR_i^\top$ and possibly an additional zero block (if the sum of their ranks is $< m$). In such a basis, the eigenvalues of $Y$ are just the multiset union of the eigenvalues of $R_iR_i^\top$, i.e., the squared singular values of each $R_i$.

Equivalently, if $\mu_1 \geq \mu_2 \geq \cdots \geq 0$ are the singular values of $\widehat{R}$ (padded by zeros when necessary), then

$$\lambda_j(Y) = \mu_j^2, \qquad j = 1, \ldots, m.$$

**Step 5: Eigenvalue monotonicity and conclusion.** From equation 10, $X \succeq Y$, i.e., $X - Y$ is positive semidefinite. By Weyl's monotonicity theorem (see Theorem 3), the eigenvalues of $X$ and $Y$ satisfy

$$\lambda_j(X) \geq \lambda_j(Y), \qquad j = 1, \ldots, m.$$

Combining this with the identities

$$\lambda_j(X) = \sigma_j^2(M_K), \qquad \lambda_j(Y) = \mu_j^2,$$

we obtain

$$\sigma_j^2(M_K) \geq \mu_j^2, \qquad j = 1, \ldots, m.$$

Since both sides are nonnegative, taking square roots preserves the inequality:

$$\sigma_j(M_K) \geq \mu_j, \qquad j = 1, \ldots, m.$$

This is exactly the claimed bound. $\qquad\square$

## C.3 Proof of Corollary 1

*Proof.* By Theorem 2, $\sigma_j(M_K) \geq \mu_j$ for all $j \geq 1$, hence

$$\sum_{j=1}^r \sigma_j^2(M_K) \geq \sum_{j=1}^r \mu_j^2.$$

The Frobenius norm of $M_K$ satisfies

$$\|M_K\|_F^2 = \sum_{j \geq 1} \sigma_j^2(M_K) = \sum_{i=1}^K \|A_i\|_F^2,$$

since $M_K$ is the horizontal concatenation of the blocks $A_i$. By Theorem 4,

$$\mathcal{E}_r^2(M_K) = \sum_{j>r} \sigma_j^2(M_K) = \|M_K\|_F^2 - \sum_{j=1}^r \sigma_j^2(M_K).$$

Substituting $\|M_K\|_F^2 = \sum_{i=1}^K \|A_i\|_F^2$ and using the lower bound on the top-$r$ energy yields

$$\mathcal{E}_r^2(M_K) = \sum_{i=1}^K \|A_i\|_F^2 - \sum_{j=1}^r \sigma_j^2(M_K) \leq \sum_{i=1}^K \|A_i\|_F^2 - \sum_{j=1}^r \mu_j^2,$$

which is exactly equation 5. $\qquad\square$

## C.4 Proof of Corollary 3

*Proof.* By definition of the Frobenius norm and the block structure of $M_K$, we have

$$\|M_K\|_F^2 = \sum_{i=1}^K \|A_i\|_F^2.$$

On the other hand, for the true singular values $\sigma_1(M_K) \geq \sigma_2(M_K) \geq \cdots$, the Eckart–Young–Mirsky theorem (Theorem 4) gives

$$\mathcal{E}_r^2(M_K) = \sum_{j>r} \sigma_j^2(M_K) = \sum_{j \geq 1} \sigma_j^2(M_K) - \sum_{j=1}^r \sigma_j^2(M_K) = \|M_K\|_F^2 - \sum_{j=1}^r \sigma_j^2(M_K).$$

The plug-in estimator $\widetilde{\mathcal{E}}_r(M_K)$ is obtained by replacing the unknown true singular values $\sigma_j(M_K)$ in this identity with their incremental approximations $\widetilde{\sigma}_j(M_K)$:

$$\widetilde{\mathcal{E}}_r^2(M_K) := \|M_K\|_F^2 - \sum_{j=1}^r \widetilde{\sigma}_j^2(M_K) = \sum_{i=1}^K \|A_i\|_F^2 - \sum_{j=1}^r \widetilde{\sigma}_j^2(M_K).$$

If the incremental scheme is run without truncation, then by Corollary 3 the approximated singular values coincide with the true ones, $\widetilde{\sigma}_j(M_K) = \sigma_j(M_K)$ for all $j$, and therefore

$$\widetilde{\mathcal{E}}_r^2(M_K) = \|M_K\|_F^2 - \sum_{j=1}^r \sigma_j^2(M_K) = \mathcal{E}_r^2(M_K),$$

which implies $\widetilde{\mathcal{E}}_r(M_K) = \mathcal{E}_r(M_K)$. In the truncated case, $\widetilde{\mathcal{E}}_r(M_K)$ is an approximation to $\mathcal{E}_r(M_K)$ obtained by this plug-in substitution. $\qquad\square$

# D    Examples

## D.1    Worst-case looseness of the Weyl-based bound

**Example 1** (Perfect compressibility with a loose Weyl bound). *Consider rank-one matrices*

$$A_j = u\, s_j\, v_j^\top, \qquad \|u\|_2 = 1,$$

*sharing the same left singular vector $u$, with arbitrary right singular vectors $v_j$ and scalars $s_j > 0$. The concatenated matrix $M = [A_1, \ldots, A_K]$ has rank one, and hence*

$$\mathcal{E}_1^2(M) = 0.$$

*However, Theorem 1 yields*

$$\mathcal{E}_1^2(M) \ \leq \ \sum_{j=1}^{K} s_j^2 \ - \ \max_j s_j^2,$$

*which grows with $K$ unless a single block dominates. Thus, even in a maximally compressible setting, the Weyl-based bound can significantly overestimate the true reconstruction error.*

## D.2    Exactness and degeneracy of the residual-based bound

**Proposition 1** (Exactness and degeneracy of the residual-based bound). *Let*

$$M_K = \begin{bmatrix} A_1 & A_2 & \cdots & A_K \end{bmatrix} \in \mathbb{R}^{m \times (n_1 + \cdots + n_K)}$$

*be formed by horizontal concatenation, and let $R_1, \ldots, R_K$ be the incremental residuals defined as in Theorem 2. Let $\mu_1 \geq \mu_2 \geq \cdots \geq 0$ denote the singular values of the concatenated residual matrix $\widehat{R} := [R_1, \ldots, R_K]$.*

***Exactness under orthogonal subspace growth.*** *If the residual subspaces $\mathrm{range}(R_1), \ldots, \mathrm{range}(R_K)$ are mutually orthogonal and*

$$\mathrm{range}(M_K) = \bigoplus_{i=1}^{K} \mathrm{range}(R_i),$$

*then the singular values of $M_K$ coincide with those of $\widehat{R}$, i.e.*

$$\sigma_j(M_K) = \mu_j \quad \text{for all } j,$$

*and the residual-based bound in Corollary 1 holds with equality for every target rank $r$.*

***Degeneracy under nested column spaces.*** *If*

$$\mathrm{range}(A_1) \supseteq \mathrm{range}(A_2) \supseteq \cdots \supseteq \mathrm{range}(A_K),$$

*then $R_i = 0$ for all $i \geq 2$, and the residual-based bound reduces to*

$$\mathcal{E}_r^2(M_K) \leq \sum_{i=2}^{K} \|A_i\|_F^2,$$

*which may be arbitrarily loose when all blocks lie in a common low-dimensional subspace.*

*Proof.* **Exactness.** Under the stated assumptions, the residual subspaces $\mathrm{range}(R_1), \ldots, \mathrm{range}(R_K)$ are mutually orthogonal and together span $\mathrm{range}(M_K)$. Consequently, the Gram matrix of the concatenated residuals satisfies

$$\widehat{R}\widehat{R}^\top = \sum_{i=1}^{K} R_i R_i^\top = M_K M_K^\top.$$

Thus the eigenvalues of $M_K M_K^\top$ coincide with those of $\widehat{R}\widehat{R}^\top$, implying $\sigma_j(M_K) = \mu_j$ for all $j$. Substituting into the definition of the optimal rank-$r$ approximation error yields equality in the bound of Corollary 1.

**Degeneracy.** If range$(A_i) \subseteq$ range$(M_{i-1})$, then by definition of the residual

$$R_i = (I - Q_{i-1}Q_{i-1}^\top)A_i = 0.$$

Under the nesting assumption, this holds for all $i \geq 2$. The claimed bound then follows immediately from Corollary 1 by noting that $\mu_j = 0$ for all $j > \text{rank}(A_1)$.

$\square$

### D.3 Underestimation of the approximate estimator under repeated truncation

**Example 2** (Underestimation via repeated truncation of a nearly aligned secondary direction)**.** *Consider the target rank $r = 1$ and let*

$$e_1 = \begin{bmatrix} 1 \\ 0 \end{bmatrix}, \qquad e_2 = \begin{bmatrix} 0 \\ 1 \end{bmatrix},$$

*and*

$$u_\theta := \cos\theta\, e_1 + \sin\theta\, e_2, \qquad 0 < \theta \ll 1.$$

*Fix amplitudes $\alpha, \beta > 0$ with $\alpha \gg \beta$, and define three rank-one blocks*

$$A_1 = \alpha e_1, \qquad A_2 = \beta u_\theta, \qquad A_3 = \beta u_\theta.$$

*Let*

$$M_3 = [A_1\ A_2\ A_3].$$

*We compare the true rank-1 error $\mathcal{E}_1(M_3)$ with the plug-in estimator $\widetilde{\mathcal{E}}_1(M_3)$ produced by the truncated incremental scheme of Corollary 2, under the processing order $(A_1, A_2, A_3)$ and with truncation back to rank 1 after each update.*

**True rank-1 error.** *The true Gram matrix is*

$$M_3 M_3^\top = \alpha^2 e_1 e_1^\top + 2\beta^2 u_\theta u_\theta^\top.$$

*In the basis $\{e_1, e_2\}$ this is*

$$\begin{bmatrix} \alpha^2 + 2\beta^2 \cos^2\theta & 2\beta^2 \sin\theta\cos\theta \\ 2\beta^2 \sin\theta\cos\theta & 2\beta^2 \sin^2\theta \end{bmatrix}.$$

*Its smaller eigenvalue equals the squared optimal rank-1 error. Since*

$$\det(M_3 M_3^\top) = 2\alpha^2\beta^2 \sin^2\theta,$$

*and the larger eigenvalue is $\alpha^2 + O(\beta^2)$, we obtain*

$$\mathcal{E}_1^2(M_3) = \lambda_{\min}(M_3 M_3^\top) = 2\beta^2 \sin^2\theta + O\left(\frac{\beta^4}{\alpha^2} \sin^2\theta\right).$$

*Thus, for $\alpha \gg \beta$ and small $\theta$, the true discarded energy is asymptotically*

$$\mathcal{E}_1^2(M_3) \approx 2\beta^2 \sin^2\theta.$$

**What the incremental estimator keeps after the first two blocks.** *After processing $A_1$ and $A_2$, the exact two-block Gram matrix is*

$$G_2 := A_1 A_1^\top + A_2 A_2^\top = \alpha^2 e_1 e_1^\top + \beta^2 u_\theta u_\theta^\top.$$

*Its top eigenpair is retained and the smaller eigenvalue is discarded by rank-1 truncation. By the same calculation as above,*

$$\lambda_{\min}(G_2) = \beta^2 \sin^2 \theta + O\left(\frac{\beta^4}{\alpha^2} \sin^2 \theta\right).$$

*Hence, after the second step, the incremental state has already discarded an amount of energy of order*

$$\beta^2 \sin^2 \theta.$$

*Let $q$ denote the retained unit eigenvector of $G_2$. Since $\alpha \gg \beta$, standard perturbation of the top eigenvector gives*

$$q = e_1 + O\left(\frac{\beta^2}{\alpha^2} \sin \theta\right),$$

*so the retained one-dimensional subspace remains very close to $\mathrm{span}(e_1)$.*

**Third update and final plug-in estimate.** *When $A_3$ is appended, the incremental scheme updates the rank-1 approximation of $G_2$ rather than the full $G_2$. Thus the matrix used internally before the final truncation is*

$$\widetilde{G}_3 = \lambda_{\max}(G_2)\, qq^\top + \beta^2 u_\theta u_\theta^\top,$$

*where the previously discarded component $\lambda_{\min}(G_2)$ is no longer present.*

*The plug-in estimator at the final step is therefore governed by the smaller eigenvalue of $\widetilde{G}_3$. Since $q$ is asymptotically aligned with $e_1$, the same calculation yields*

$$\lambda_{\min}(\widetilde{G}_3) = \beta^2 \sin^2 \theta + O\left(\frac{\beta^4}{\alpha^2} \sin^2 \theta\right).$$

*Therefore*

$$\widetilde{\mathcal{E}}_1^2(M_3) = \beta^2 \sin^2 \theta + O\left(\frac{\beta^4}{\alpha^2} \sin^2 \theta\right).$$

**Comparison.** *Combining the two expansions, we obtain*

$$\mathcal{E}_1^2(M_3) = 2\beta^2 \sin^2 \theta + O\left(\frac{\beta^4}{\alpha^2} \sin^2 \theta\right), \qquad \widetilde{\mathcal{E}}_1^2(M_3) = \beta^2 \sin^2 \theta + O\left(\frac{\beta^4}{\alpha^2} \sin^2 \theta\right).$$

*Hence, for sufficiently large $\alpha/\beta$ and sufficiently small $\theta$,*

$$\widetilde{\mathcal{E}}_1(M_3) < \mathcal{E}_1(M_3).$$

**Interpretation.** *The key point is that the direction $u_\theta$ is not lost forever: it is reintroduced through the residual at the third step. However, because the algorithm truncates back to rank 1 after processing $A_2$, the secondary spectral component created by $A_2$ is discarded once, and the later update with $A_3$ cannot recover this previously discarded contribution exactly. The resulting plug-in estimator therefore captures only the newly added* off-subspace *energy at the final step, but not the portion that was already lost at the previous truncation.*

---

**Algorithm 1** Weyl-based max-norm clustering

---

**Require:** Blocks $\{A_j\}_{j=1}^K$, tolerance $\varepsilon$, width budget $r_{\text{target}}$
**Ensure:** Clusters $\mathcal{C}$
 1: Sort blocks by decreasing Frobenius norm
 2: **while** blocks remain **do**
 3:     Choose the largest norm remaining block as anchor
 4:     Form a head by concatenating the largest norm blocks up to width $r_{\text{target}}$
 5:     Add smallest norm remaining blocks while relative tail energy $\leq \varepsilon$
 6:     Output the resulting cluster and remove its blocks
 7: **end while**

---

# E    Algorithms

## E.1    Fast Weyl-based clustering

Here we introduce a lightweight clustering heuristic derived from the simplified Weyl-type bound in Theorem 1. The key idea is to form clusters around a dominant block and to allow additional blocks to join only if their contribution does not violate the prescribed error tolerance.

The user specifies a tolerance $\varepsilon > 0$ controlling the admissible *relative* rank-$r$ approximation error within each cluster. Assuming the target rank satisfies $r \geq \max_{j \in C} \text{rank}(A_j)$, the bound guarantees that every cluster $C$ produced by Algorithm 1 obeys

$$\frac{\mathcal{E}_r(M_C)}{\|M_C\|_F} \leq \varepsilon.$$

**Basic idea.**    The Weyl-based bound depends only on the Frobenius norms of the blocks and is dominated by the largest block in the cluster. This leads to an *anchor-based* clustering strategy: each cluster is initialized by a block with large Frobenius norm, and additional blocks are appended as long as the resulting upper bound on the approximation error remains below the tolerance.

**Merge certificate.**    Let $C$ be a current cluster and let $A_j$ be a candidate block. Using the Weyl-based bound, we test whether adding $A_j$ preserves the error constraint, i.e., whether

$$\frac{\widehat{\mathcal{E}}_r(M_{C \cup \{j\}})}{\|M_{C \cup \{j\}}\|_F} \leq \varepsilon,$$

where the estimator depends only on the Frobenius norms of the blocks and the maximum norm within the cluster. This condition serves as a *merge certificate*.

**Algorithmic structure.**    The blocks are first sorted in descending order of their Frobenius norms. This ensures that each cluster is anchored by its largest block, which stabilizes the Weyl bound and avoids overly conservative estimates. The algorithm then proceeds greedily: starting from the largest unassigned block, it forms a new cluster and iteratively adds the next *smallest norm* blocks as long as the merge criterion is satisfied.

The overall complexity is $O(K \log K)$, dominated by sorting the block norms, while all clustering decisions require only scalar operations. In Section 4 we show that, despite its simplicity and reliance on coarse norm information, this Weyl-based clustering already yields strong compression performance in practice.

**Step-by-step procedure.**    The greedy clustering procedure can be summarized as follows:

1. Compute the Frobenius norms $\|A_j\|_F$ for all blocks and sort the blocks in descending order.

2. Initialize a new cluster with the largest unassigned block $A_{j_0}$. Set

$$S_C = \|A_{j_0}\|_F^2, \qquad m_C = \|A_{j_0}\|_F^2,$$

---

**Algorithm 2** Residual-based clustering

---

**Require:** Blocks $\{A_j\}_{j=1}^K$, tolerance $\varepsilon$, target rank $r$
**Ensure:** Clusters $\mathcal{C}$
1: Sort blocks by decreasing $\|A_j\|_F$
2: **while** blocks remain **do**
3:     Select the largest remaining block as anchor
4:     Maintain an incremental rank-$r$ energy estimate for the cluster
5:     Add remaining blocks (by small norm or small residual) while

$$\frac{\|M_C\|_F^2 - \sum_{\ell=1}^r \mu_\ell^2}{\|M_C\|_F^2} \leq \varepsilon^2$$

6:     Output the cluster and remove its blocks
7: **end while**

---

where $S_C = \sum_{i \in C} \|A_i\|_F^2$ denotes the cumulative Frobenius energy of the cluster and $m_C = \max_{i \in C} \|A_i\|_F^2$ denotes the maximum block energy within the cluster.

3. Form the set of remaining unassigned blocks and process them in *increasing* order of their Frobenius norms.

4. For each candidate block $A_j$:

   (a) compute the updated cumulative energy

   $$S_{C \cup \{j\}} = S_C + \|A_j\|_F^2;$$

   (b) update the maximum block energy

   $$m_{C \cup \{j\}} = \max\big(m_C, \|A_j\|_F^2\big);$$

   (c) evaluate the Weyl-based upper bound on the rank-$r$ approximation error for the enlarged cluster $C \cup \{j\}$ using the norm-only quantities $S_{C \cup \{j\}}$ and $m_{C \cup \{j\}}$;

   (d) accept the block if the corresponding relative error satisfies

   $$\frac{\widehat{\mathcal{E}}_r(M_{C \cup \{j\}})}{\|M_{C \cup \{j\}}\|_F} \leq \varepsilon,$$

   in which case update $S_C \leftarrow S_{C \cup \{j\}}$ and $m_C \leftarrow m_{C \cup \{j\}}$;

   (e) otherwise reject the block for this cluster and leave it available for future clusters.

5. Repeat until no further block can be added without violating the error constraint. Then initialize a new cluster from the next largest unassigned block and continue until all blocks are assigned.

## E.2 Residual-based clustering

Here we introduce a clustering procedure derived from the residual-based bound of Corollary 1. In contrast to the Weyl-based algorithm, which evaluates a candidate merge using only block energies, the present method explicitly tracks how much *new subspace structure* each candidate block contributes beyond what is already represented by the current cluster. This leads to a more expensive but substantially tighter clustering criterion. The resulting procedure is still greedy, but its decisions are driven by a more informative spectral surrogate. This is precisely the setting in which the residual-based theory is useful: it turns the qualitative notion of "does this block bring genuinely new directions?" into a computable merge certificate.

**Basic idea.** Suppose that a cluster $C$ has already been formed, and let

$$M_C := [A_j]_{j \in C}$$

be its concatenated matrix. The central object maintained by the algorithm is an orthonormal basis $Q_C$ for the current cluster subspace. Intuitively, $Q_C$ represents the directions already explained by the blocks that have been assigned to $C$. When a new candidate block $A_j$ is considered, we do not immediately ask whether its Frobenius norm is small. Instead, we first ask a more relevant question: *how much of $A_j$ lies outside the subspace already represented by the cluster?*

This is measured by the residual

$$R_j := (I - Q_C Q_C^\top) A_j.$$

If $R_j$ is small, then most of $A_j$ is already aligned with the current cluster and can potentially be absorbed with little additional approximation error. If $R_j$ is large, then $A_j$ introduces substantial new directions and is therefore less compatible with the existing cluster.

**Why residuals matter.** The residual-based bound differs fundamentally from the Weyl-based one. The Weyl bound treats the cluster as being controlled by a single dominant block, and therefore ignores how multiple blocks may jointly contribute useful shared structure. By contrast, the residual formulation accumulates the innovations introduced by all accepted blocks. Each time a new block contributes a component outside the span of the current cluster, that contribution is stored through the residual representation and enters the singular values used in the bound. In this sense, the algorithm does not merely track which block is largest; it tracks how the *cluster subspace evolves* as blocks are added.

**Merge certificate.** For a current cluster $C$, let

$$\mu_1(M_C) \geq \mu_2(M_C) \geq \cdots$$

denote the singular values of the concatenated residual matrix associated with the blocks in $C$. Corollary 1 yields the bound

$$\mathcal{E}_r^2(M_C) \leq \sum_{j \in C} \|A_j\|_F^2 - \sum_{i=1}^r \mu_i^2(M_C).$$

This expression is the key algorithmic primitive. It provides a *merge certificate*: if, after tentatively adding a candidate block $A_j$, the resulting upper bound remains below the prescribed error threshold, then the merge is accepted; otherwise it is rejected.

Equivalently, when testing whether $A_j$ may join the current cluster, the algorithm performs the following conceptual check:

1. update the residual representation as if $A_j$ were appended to $C$;

2. recompute the leading residual singular values for the enlarged cluster;

3. evaluate the residual-based upper bound;

4. accept the merge only if the corresponding *relative* error estimate does not exceed the tolerance $\varepsilon$.

Thus, the theory is used directly: the bound is not just an analytical result, but the actual decision rule governing cluster formation.

**What is maintained during clustering.** The algorithm maintains, for each active cluster, the following quantities:

- the set of assigned block indices $C$;

- the orthonormal basis $Q_C$ representing the current cluster subspace;

- the cumulative Frobenius energy

$$S_C := \sum_{j \in C} \|A_j\|_F^2;$$

- the leading singular values of the concatenated residual representation used in the merge certificate.

Each accepted block changes both the total energy $S_C$ and the residual spectrum. The total energy increases by $\|A_j\|_F^2$, while the residual spectrum changes according to the newly introduced directions in $R_j$. This is why the method is more informative than the Weyl-based clustering: it keeps track of *how* the new energy is distributed relative to the existing cluster geometry, not just *how much* total energy is added.

**Candidate ordering.** At each step, candidate blocks are considered according to a user-selected `sort_mode`. This ordering does not change the validity of the bound, but it changes the order in which the greedy procedure explores feasible merges, and can therefore affect the final clustering outcome.

Two modes are supported:

- `frobenius`: candidates are ordered by increasing Frobenius norm. This is a computationally cheap heuristic. It prioritizes blocks with small total energy, which are often easier to absorb into an existing cluster. This mode is conceptually close to the max-norm / Weyl-based strategy, but uses the sharper residual-based certificate when deciding whether a merge is actually valid.

- `residual`: candidates are ordered by increasing residual norm

$$\|(I - Q_C Q_C^\top)A_j\|_F.$$

This mode is more geometry-aware. Instead of preferring blocks that are merely small in norm, it prefers blocks that add the least *new* information relative to the current cluster. In practice, this often yields tighter and more semantically coherent clusters, but requires recomputing candidate residuals with respect to the evolving basis $Q_C$.

**Step-by-step procedure.** The greedy clustering procedure can be summarized as follows:

1. Select the unassigned block with the largest Frobenius norm and use it to initialize a new cluster, denoted $A_{j_0}$. Construct an orthonormal basis $Q_C$ for its column space (or its leading rank-$r$ subspace), initialize the cumulative Frobenius energy

$$S_C = \|A_{j_0}\|_F^2,$$

and initialize the residual representation.

2. Form the list of remaining candidate blocks not yet assigned to any cluster and order them according to the selected `sort_mode`.

3. For each candidate block $A_j$:

   (a) compute its residual with respect to the current cluster subspace

   $$R_j = (I - Q_C Q_C^\top)A_j;$$

   (b) tentatively update the residual representation of the cluster by incorporating $R_j$, and update the leading residual singular values $\{\mu_i\}$ for the enlarged cluster;

   (c) update the cumulative energy

   $$S_{C \cup \{j\}} = S_C + \|A_j\|_F^2;$$

   (d) evaluate the residual-based upper bound

   $$\widehat{\mathcal{E}}_r^2(M_{C \cup \{j\}}) = S_{C \cup \{j\}} - \sum_{i=1}^r \mu_i^2;$$

---

**Algorithm 3** Approximate residual-based clustering

---

**Require:** Blocks $\{A_j\}_{j=1}^K$, tolerance $\varepsilon$, target rank $r$
**Ensure:** Clusters $\mathcal{C}$
1: Sort blocks by decreasing $\|A_j\|_F$
2: **while** blocks remain **do**
3:     Select the largest remaining block as anchor
4:     Initialize a cluster subspace model that tracks the leading $r$ singular values approximately
5:     Add remaining blocks (by small norm or small residual) while

$$\frac{\|M_C\|_F^2 - \sum_{\ell=1}^r \widetilde{\sigma}_\ell^2(M_C)}{\|M_C\|_F^2} \leq \varepsilon^2$$

    where $\sum_{\ell=1}^r \widetilde{\sigma}_\ell^2(M_C)$ is an incremental approximation
6:     Output the cluster and remove its blocks
7: **end while**

---

    (e) accept the block if the corresponding relative error satisfies

$$\frac{\widehat{\mathcal{E}}_r(M_{C\cup\{j\}})}{\|M_{C\cup\{j\}}\|_F} \leq \varepsilon,$$

    in which case update $Q_C$, $S_C$, and the residual spectrum;
    (f) otherwise reject the block for this cluster and leave it available for future clusters.

4. Repeat until no further candidate can be added without violating the error constraint. Then initialize a new cluster from the remaining unassigned block with the largest Frobenius norm and continue until all blocks are assigned.

**Cost and trade-off.** Compared to the Weyl-based Algorithm 1, the present approach is more expensive because it must repeatedly project candidate blocks onto the orthogonal complement of the current cluster subspace and update the residual singular values. The benefit of this extra cost is a much tighter characterization of the achievable rank-$r$ approximation error. Empirically, this allows the residual-based algorithm to identify feasible merges that the coarser Weyl-based method rejects, especially when compression arises from shared structure distributed across multiple blocks rather than from dominance of a single anchor block.

### E.3 Approximate clustering via incremental truncated SVD

We next introduce a scalable clustering procedure that replaces the residual-based merge certificate with a fast plug-in estimator $\widetilde{\mathcal{E}}_r(\cdot)$ obtained from an incremental truncated SVD. The key idea is to approximate the dominant rank-$r$ subspace of each cluster on the fly and to use the energy captured by this subspace to estimate the compression error, without explicitly forming residuals or computing large-scale singular value decompositions.

**Basic idea.** As in Algorithm 2, the clustering process is greedy: clusters are built incrementally by appending blocks as long as a merge criterion is satisfied. The crucial difference lies in how the approximation error is estimated. Instead of tracking the residual spectrum, the algorithm maintains a compressed rank-$r$ representation of the current cluster matrix $M_C$ via an incremental truncated SVD. This representation consists of a basis $Q_C$ together with a small set of singular values that approximate the leading spectrum of $M_C$.

Intuitively, $Q_C$ represents the directions currently captured by the cluster, while the associated singular values encode how much energy is retained in these directions. The estimator $\widetilde{\mathcal{E}}_r(M_C)$ is then obtained by comparing the total Frobenius energy of the cluster with the energy captured by this rank-$r$ approximation.

**Merge certificate.** Given a current cluster $C$ and a candidate block $A_j$, the algorithm proceeds as follows. First, it performs a tentative incremental SVD update of the cluster representation as if $A_j$ were appended to $C$, maintaining only the leading $r$ singular directions. This yields an updated approximation of the dominant subspace and its associated singular values. The estimated rank-$r$ approximation error is then computed as

$$\widetilde{\mathcal{E}}_r^2(M_{C \cup \{j\}}) \;=\; \sum_{i \in C \cup \{j\}} \|A_i\|_F^2 \;-\; \sum_{k=1}^{r} \widetilde{\sigma}_k^2,$$

where $\widetilde{\sigma}_k$ are the singular values produced by the incremental procedure. The block $A_j$ is accepted if the corresponding relative error satisfies

$$\frac{\widetilde{\mathcal{E}}_r(M_{C \cup \{j\}})}{\|M_{C \cup \{j\}}\|_F} \;\leq\; \varepsilon.$$

Thus, as in the exact methods, clustering is driven by a merge certificate, but here the certificate is computed approximately.

**What is maintained during clustering.** For each cluster, the algorithm maintains:

- a rank-$r$ orthonormal basis $Q_C$ representing the current approximation of the cluster subspace;

- a set of approximate singular values $\{\widetilde{\sigma}_k\}_{k=1}^{r}$;

- the cumulative Frobenius energy
$$S_C := \sum_{j \in C} \|A_j\|_F^2.$$

Each merge step updates these quantities using only matrices of size at most $O(r)$, which makes the method independent of the number of blocks already assigned to the cluster.

**Candidate ordering.** As in Algorithm 2, two ordering strategies are supported:

- `frobenius`: candidates are processed in increasing order of $\|A_j\|_F$;

- `residual`: candidates are ordered by the norm of the projected residual $(I - Q_C Q_C^\top)A_j$, prioritizing blocks that are well aligned with the current approximate subspace.

**Step-by-step procedure.** The greedy clustering procedure can be summarized as follows:

1. Select the unassigned block with the largest Frobenius norm and use it to initialize a new cluster, denoted $A_{j_0}$. Compute its rank-$r$ SVD to obtain an orthonormal basis $Q_C$ and singular values $\{\widetilde{\sigma}_k\}_{k=1}^{r}$. Initialize the cumulative Frobenius energy
$$S_C = \|A_{j_0}\|_F^2.$$

2. Form the list of remaining candidate blocks not yet assigned to any cluster and order them according to the selected `sort_mode`.

3. For each candidate block $A_j$:

   (a) form the projection of $A_j$ onto the current subspace and its orthogonal complement:
   $$A_j^{\|} = Q_C Q_C^\top A_j, \qquad A_j^{\perp} = (I - Q_C Q_C^\top)A_j;$$

   (b) construct a small augmented matrix representing the current low-rank approximation together with the new block, and perform a truncated SVD update to obtain updated singular values $\{\widetilde{\sigma}_k^{\text{new}}\}_{k=1}^{r}$ and an updated basis $Q_C^{\text{new}}$;

(c) update the cumulative energy

$$S_{C \cup \{j\}} = S_C + \|A_j\|_F^2;$$

(d) evaluate the plug-in estimate of the rank-$r$ approximation error

$$\widetilde{\mathcal{E}}_r^2(M_{C \cup \{j\}}) = S_{C \cup \{j\}} - \sum_{k=1}^{r} (\widetilde{\sigma}_k^{\text{new}})^2;$$

(e) accept the block if the corresponding relative error satisfies

$$\frac{\widetilde{\mathcal{E}}_r(M_{C \cup \{j\}})}{\|M_{C \cup \{j\}}\|_F} \le \varepsilon,$$

in which case update

$$Q_C \leftarrow Q_C^{\text{new}}, \quad \{\widetilde{\sigma}_k\} \leftarrow \{\widetilde{\sigma}_k^{\text{new}}\}, \quad S_C \leftarrow S_{C \cup \{j\}};$$

(f) otherwise reject the block for this cluster and leave it available for future clusters.

4. Repeat until no further candidate can be added without violating the tolerance. Then initialize a new cluster from the remaining unassigned block with the largest Frobenius norm and continue until all blocks are assigned.

**Cost and scalability.** Each update operates only on small matrices of size proportional to $r$, independent of the cluster size. This makes the method particularly suitable for large-scale settings where forming the full concatenated matrix or tracking exact residual spectra is prohibitively expensive.

**Limitations.** Unlike Algorithms 1 and 2, this approach does not provide a formal worst-case bound on the true compression error. The approximation relies on repeated truncation of the evolving subspace: directions that are weak when first observed may be discarded and later reappear with significant energy. As a result, the estimator may underestimate or overestimate the true error, and the outcome may depend on the order in which blocks are processed. Nevertheless, as demonstrated in Section 4, the estimator is empirically accurate enough to guide clustering decisions and achieves the best trade-off between computational efficiency and compression quality in large-scale regimes.

## F   Additional Slack Diagnostics Across Datasets

In this section we provide additional diagnostics of estimator conservativeness across all datasets considered in Table 1. The goal is to further analyze the behavior of the predicted reconstruction error $\widetilde{\mathcal{E}}_r$ relative to the true truncated SVD error $\mathcal{E}_r$, via the slack

$$\Delta = \widetilde{\mathcal{E}}_r - \mathcal{E}_r.$$

**Experimental setup.** The experiments follow the same protocol as in the main paper. For each dataset (Qualcomm MIMO, BigEarthNet, and PDEBench), we sample clusters of increasing size and evaluate all estimators on 10 independent trials per configuration, showing *all individual outcomes*. Clustering is performed under a fixed relative reconstruction error constraint of 5%. The target rank is fixed to $r = 20$ for Qualcomm and BigEarthNet, and $r = 67$ for PDEBench. These settings exactly match those used in Table 2, ensuring direct comparability between compression results and estimator diagnostics.

**Non-negativity of slack for approximate estimator.** Across all datasets and all experimental configurations, we did not observe any instances of negative slack for the approximate incremental estimator. That is, the plug-in estimator consistently *overestimates* or closely matches the true reconstruction error in practice. While the estimator is not theoretically guaranteed to be an upper bound, these results indicate that under realistic data distributions it behaves as a conservative surrogate.

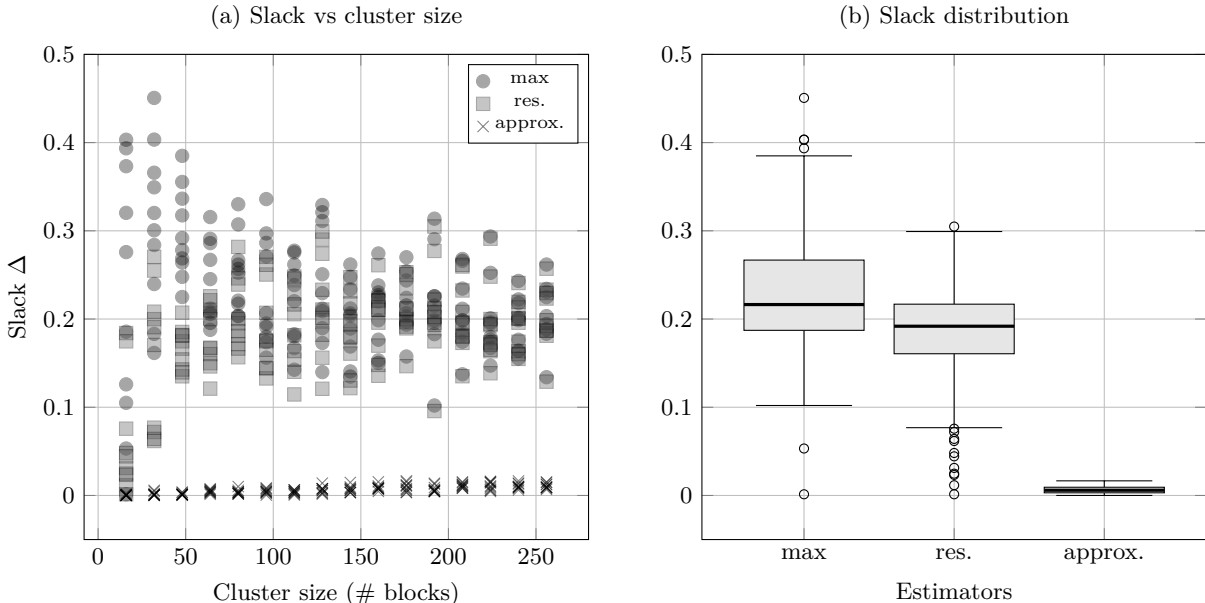

Figure 3: Qualcomm MIMO diagnostic of estimator conservativeness. Slack $\Delta = \widetilde{\mathcal{E}}_r - \mathcal{E}_r$ between predicted and true rank-$r$ SVD reconstruction error. For each cluster size and estimator, all 10 independent trials (uniform block samples) are shown. (a) slack versus cluster size; (b) empirical slack distribution across estimators.

**Qualcomm MIMO.** Figure 3 shows that the residual-based estimator consistently produces tighter predictions than the max-norm bound. This is reflected both in the scatter plots and in the distribution of slack values, where residual clustering yields systematically smaller $\Delta$. This confirms that accounting for subspace innovation across blocks provides a more accurate characterization of the retained spectral energy than relying solely on the dominant block. The approximate estimator is significantly tighter than both exact bounds, with slack concentrated near zero.

**BigEarthNet.** Figure 4 exhibits a different regime. Both max-norm and residual-based estimators remain *highly conservative*, with large slack values and only marginal differences between them. This indicates that, in this dataset, inter-block correlations are not sufficiently captured by the residual construction to substantially tighten the bound. As a result, both exact estimators lead to similarly restrictive clustering decisions. In contrast, the approximate estimator remains tightly concentrated and significantly closer to the true error, enabling more effective compression.

**PDEBench.** Figure 5 highlights a regime where the exact bounds become excessively conservative. Both max-norm and residual-based estimators produce large slack values, indicating that they substantially overestimate the reconstruction error. As a result, these bounds are too restrictive to enable effective clustering under the target error constraint, which explains why exact methods fail to achieve meaningful compression in Table 2. At the same time, the approximate estimator remains tight and stable, allowing substantially larger clusters to be formed while still respecting the error constraint in practice.

# G Downstream Task Evaluation on PDEBench

To complement the reconstruction-error-based evaluation, we perform a downstream task analysis on the PDEBench dataset. While our primary objective is to control low-rank approximation error, it is important to understand how this error translates to performance in practical predictive settings.

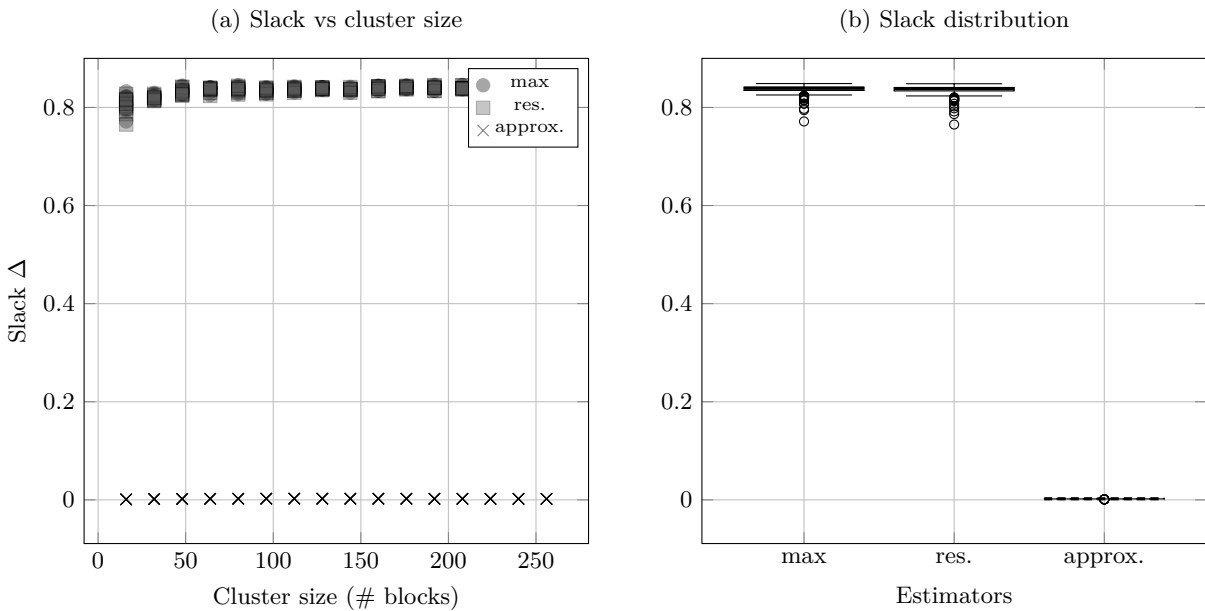

Figure 4: BigEarthNet diagnostic of estimator conservativeness. Slack $\Delta = \widetilde{\mathcal{E}}_r - \mathcal{E}_r$ between predicted and true rank-$r$ SVD reconstruction error. For each cluster size and estimator, all 10 independent trials (uniform block samples) are shown. (a) slack versus cluster size; (b) empirical slack distribution across estimators.

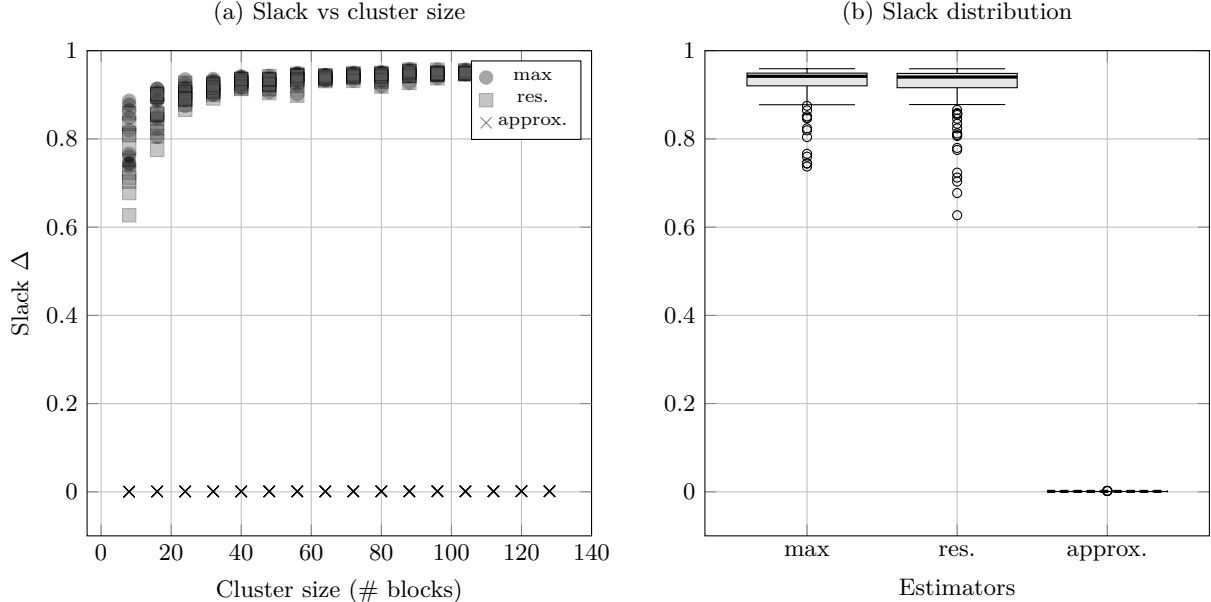

Figure 5: PDEBench diagnostic of estimator conservativeness. Slack $\Delta = \widetilde{\mathcal{E}}_r - \mathcal{E}_r$ between predicted and true rank-$r$ SVD reconstruction error. For each cluster size and estimator, all 10 independent trials (uniform block samples) are shown. (a) slack versus cluster size; (b) empirical slack distribution across estimators.

We consider a simple time-series forecasting task on the PDEBench dataset. Given a single input timestep, the goal is to predict the system state 20 steps ahead. We use a fixed training pipeline across all methods and vary only the representation of the input data via compression.

The following methods are compared:

- **No compression (baseline):** original data without dimensionality reduction.

- **Full concatenation + SVD:** a single SVD applied to the fully concatenated matrix.

- **Approximate residual clustering (ours):** clustering using the approximate residual-based estimator followed by per-cluster truncated SVD, with varying target error thresholds $\varepsilon$.

We report mean squared error (MSE), relative $\ell_2$ error, and $R^2$ score on a held-out test set. All experiments are conducted with fixed rank $r = 10$ and horizon 20.

| Method | $\varepsilon$ | Comp. ↑ | Recon. ↓ | MSE ↓ | Rel-$\ell_2$ ↓ | $R^2$ ↑ |
|---|---|---|---|---|---|---|
| Baseline (no compression) | – | 1.0 | 0 | $2.66 \cdot 10^{-7}$ | $6.64 \cdot 10^{-4}$ | 0.9999996 |
| Full SVD (all blocks) | – | 20.0 | $1.15 \cdot 10^{-1}$ | $7.21 \cdot 10^{1}$ | $1.11 \cdot 10^{1}$ | $-121.5$ |
| Ours (approx. residual) | 0.005 | 3.59 | $2.48 \cdot 10^{-3}$ | $8.80 \cdot 10^{-7}$ | $1.21 \cdot 10^{-3}$ | 0.9999985 |
| Ours (approx. residual) | 0.01 | 4.95 | $5.28 \cdot 10^{-3}$ | $9.93 \cdot 10^{-6}$ | $4.06 \cdot 10^{-3}$ | 0.9999834 |
| Ours (approx. residual) | 0.02 | 6.09 | $9.79 \cdot 10^{-3}$ | $5.12 \cdot 10^{-5}$ | $9.23 \cdot 10^{-3}$ | 0.9999146 |
| Ours (approx. residual) | 0.03 | 6.89 | $1.15 \cdot 10^{-2}$ | $7.91 \cdot 10^{-2}$ | $3.63 \cdot 10^{-1}$ | 0.868 |
| Ours (approx. residual) | 0.04 | 7.93 | $1.43 \cdot 10^{-2}$ | 1.20 | 1.41 | $-1.01$ |
| Ours (approx. residual) | 0.05 | 7.93 | $2.91 \cdot 10^{-2}$ | 3.30 | 2.34 | $-4.51$ |

Table 4: Downstream forecasting performance on PDEBench. Compression ratio (Comp.), reconstruction error (Recon.), and predictive metrics are reported.

Several important observations emerge from the results (see Table 4).

**(1) Reconstruction error strongly correlates with downstream performance in the low-error regime.** For small approximation error (e.g., $\varepsilon \leq 0.02$), the degradation in downstream metrics remains minimal, despite achieving up to $6\times$ compression. This suggests that controlling the Frobenius reconstruction error is sufficient to preserve predictive structure in this regime.

**(2) Sharp phase transition in performance degradation.** As the reconstruction error increases beyond a threshold (between $\varepsilon = 0.02$ and $\varepsilon = 0.03$), downstream performance deteriorates rapidly. This behavior indicates that the relationship between reconstruction error and task performance is highly non-linear.

**(3) Naive full concatenation is unstable for downstream tasks.** Despite achieving a high compression ratio ($20\times$), applying SVD to the fully concatenated matrix results in catastrophic degradation of predictive performance. This highlights the importance of clustering prior to compression, as naive concatenation destroys task-relevant structure.

**(4) Trade-off between compression and task fidelity.** The proposed method enables explicit control of this trade-off via the target error threshold $\varepsilon$. In practice, moderate compression ($4\times$–$6\times$) can be achieved with negligible downstream degradation.

**Limitations and conclusion.** We emphasize that downstream performance depends on the specific task and model used. The considered forecasting setup is intentionally simple, and more complex tasks (e.g., operator learning or control) may exhibit different sensitivity to approximation error. A comprehensive evaluation across diverse downstream tasks is left for future work.

Overall, these results support the use of reconstruction-error-based guarantees as a practical proxy for downstream performance, while also highlighting the existence of sharp degradation regimes that must be avoided in practice.

