# OpenReview forum: "Concatenated Matrix SVD: Compression Bounds, Incremental Approximation, and Error-Constrained Clustering"
_TMLR — Accepted by TMLR_

### Review · Reviewer_HNhH · 2026-02-24

**Summary Of Contributions:**

The paper explores a novel problem setting and provides theoretical derivations.

Weaknesses.

1. The presentation needs significant improvement. I found Algorithms 2 and 3 difficult to follow, and the subsection on “interpretation for clustering” is unclear. In particular, I do not understand how the theoretical bounds are used to motivate the algorithmic design. Part of this difficulty may stem from the lack of intuitive explanations of the algorithms themselves. The authors should provide clearer descriptions, step-by-step intuition, and concrete examples to help readers understand the proposed methods.

2. (Motivations.) The paper does not clearly justify why this problem is important in practice. Could the authors provide at least one concrete application scenario that can be directly formulated under their proposed problem setup?

3.  From my understanding, the bounds appear most meaningful when one matrix significantly dominates the others. However, in such cases, it is unclear why clustering is needed at all. The authors should clarify how their method behaves in more balanced regimes and why clustering remains beneficial.

**Audience:**

No

**Audience Explanation:**

The motivation is not sufficiently clarified, and the practical applicability of the proposed algorithm appears limited. Please refer to Weaknesses 2 and 3 for detailed concerns.

**Claims And Evidence:**

Yes

**Claims Explanation:**

Both theoretical and numerical results are provided. Although I did not check the full proof, the results seem reasonable to me.

**Requested Changes:**

1. Rewrite the Introduction to clearly state the motivation and practical use cases.

2. Rewrite the Related Work to justify why clustering before compression is necessary, rather than performing compression alone.

3. Add clearer explanations of the theoretical insights and explicitly connect them to the proposed algorithms.

4. Clearly identify and explain the technical challenges in the theoretical proofs; currently, these are not evident.

5. Provide more detailed and concrete algorithm descriptions, including step-by-step procedures.

---

> ### Author Response · Authors · 2026-03-29
>
> We thank the reviewer for the detailed and constructive feedback. We agree that clarity of presentation and motivation are crucial, and we have substantially revised the manuscript to address these concerns. Below we respond to each point in the order raised.
>
> ---
>
> ## Weaknesses
>
> ### (1) Algorithms hard to follow
>
> We thank the reviewer for this feedback and agree that the original presentation lacked clarity.
>
> We have revised the paper to:
> - **clarify bounds as merge criteria**, explicitly showing how each method defines a decision rule (residuals as new directions; approximate method via captured energy);
> - **connect theory to algorithms** via the pipeline
>   $\text{bound} \rightarrow \text{merge rule} \rightarrow \text{greedy clustering}$;
> - **expand algorithm descriptions** (Appendix E) with step-by-step procedures, maintained quantities, and merge logic;
> - **improve intuition**, interpreting residuals as new directions and incremental SVD as tracking a compressed subspace.
>
> ---
>
> ### (2) Motivation and practical importance
>
> We thank the reviewer for highlighting this issue.
>
> We substantially rewrote the introduction to clearly motivate the problem and provide concrete application scenarios. In particular, we now:
> - describe SVD as a standard compression primitive and explain how concatenation arises naturally for collections of matrices;
> - provide concrete use cases (e.g., neural network compression, scientific simulations, multi-view representations);
> - explicitly compare three strategies: independent compression, full concatenation, and clustering before compression;
> - introduce a clear **failure mode**: naive concatenation can increase effective rank and degrade approximation quality;
> - formulate the core problem as selecting which matrices can be safely concatenated under an error constraint.
>
> These changes make the practical relevance and problem formulation explicit.
>
> ---
>
> ### (3) Only meaningful when one matrix dominates
>
> We thank the reviewer for this insightful comment.
>
> This observation mainly applies to the **max-norm method**, which behaves as a conservative “single-anchor” rule, but does not reflect the residual-based and approximate methods.
>
> - The **residual-based method** accumulates new directions, capturing **multi-block subspace interactions**.
> - The **approximate method** tracks a shared low-rank subspace and detects structure even without a dominant matrix.
>
> In **balanced regimes**, clustering exploits shared structure that independent compression misses, while full concatenation may violate error constraints. Empirically, this is reflected in higher compression achieved by residual-based and approximate methods compared to max-norm.
>
> ---
>
> ## Requested Changes
>
> ### (1) Rewrite Introduction
>
> We have rewritten the introduction to:
> - clearly state the practical problem and its relevance,
> - provide concrete application scenarios,
> - highlight the trade-off between independent compression and full concatenation,
> - and introduce clustering as the mechanism for resolving this trade-off.
>
> ---
>
> ### (2) Rewrite Related Work
>
> We revised the related work section to explicitly justify why clustering is necessary.
>
> We now:
> - contrast independent compression vs. full concatenation,
> - highlight failure modes of heuristic grouping,
> - and position clustering as a **necessary step** for controlling reconstruction error in concatenated SVD.
>
> ---
>
> ### (3) Connect theoretical insights to algorithms
>
> We added explicit connections between theory and algorithms throughout the paper, showing how each bound directly induces a merge criterion and corresponding clustering procedure.
>
> ---
>
> ### (4) Explain technical challenges in proofs
>
> We added a dedicated paragraph (“Technical challenges”) at the end of Section 3, highlighting:
> - non-decomposability of singular values under concatenation,
> - the need for bounds computable without forming concatenated matrices,
> - and the combinatorial nature of clustering requiring efficient merge certificates.
>
> ---
>
> ### (5) More detailed algorithm descriptions
>
> We expanded the algorithm descriptions (Appendix E) to include:
> - step-by-step procedures,
> - maintained quantities,
> - merge evaluation logic,
> - and ordering strategies.
>
> These additions make the algorithms significantly more accessible and reproducible.
>
> ---
>
> ## Summary
>
> In summary, we have:
> - improved the clarity and intuition of the algorithms,
> - strengthened the motivation and practical grounding,
> - clarified behavior in balanced regimes,
> - explicitly connected theory to algorithm design,
> - and expanded technical explanations and algorithm descriptions.
>
> We thank the reviewer again for the valuable feedback, which helped substantially improve the presentation and clarity of the paper.

---

### Review · Reviewer_13qU · 2026-02-27

**Summary Of Contributions:**

This paper studies systematic approach of SVD for horizontally concatenated matrices. Given $A_1,\ldots,A_N$ where $A_j\in \mathbb{R}^{m\times n_j}$ for $j\in [N]$, the goal is to compute a partition $C_1,\ldots,C_K$ of $[N]$ and a per-partition rank $r_c$ for $c\in [K]$ so that if one computes SVD and keep the top-$r_c$ singular subspace for matrices within each partition, the sum of square of tail singular values within each cluster is minimized. The main contribution is several approaches to provide an upper bound on the optimal approximation error. The simplest approach only requires to compute the largest Frobenius norm / sum of top squared singular values based on Weyl's inequality. The second bound is via viewing the cluster as iteratively adding new blocks, and collecting the "residual" by looking at the projection of the new block onto the complement space of the cluster. Collecting these residuals and compute the singular values yields meaningful and tighter bounds for the optimal approximation error. These approaches give useful way to merge the clusters by iteratively adding blocks into another cluster. Evaluations are performed on the effectiveness of these methods, and one could see that the faster approach gives more crude approximation, while the slower one usually has better quality.

**Audience:**

Yes

**Audience Explanation:**

Concatenated SVD is a widely used approach in many fields of machine learning, importantly it's a heavily exploited heuristics, this paper provides some meaningful theoretical guidance on its effectiveness.

**Claims And Evidence:**

Yes

**Claims Explanation:**

The major portion of the paper is to prove the Weyl-based bound and residual-based clustering bound gives desired error guarantees. Proofs are provided for these statements. Experiments are also performed to show the tradeoff of these two approaches.

**Requested Changes:**

This paper provides some preliminary and useful ways for concatenated SVD, which is appreciated. On the other hand, the theoretical results are also very fundamental --- the Weyl-based approach is a simple application of Weyl's inequality, and the residual-based approach is also straightforward to prove. These results feel more like "filling the gap" for a problem that lacks study before, rather than developing some interesting new insights. Hence, I would say the theoretical contribution of this paper is quite thin. Regarding how to improve this part: I could think of two possible directions:

1. Give results other than Frobenius norm. If one is interested in spectral norm instead, what would be a good approach?

2. Give faster algorithms with end-to-end guarantees, for example, if one uses random sampling / sketching for approximate truncated SVD, what would be the end-to-end error guarantees, and runtime?

I feel the empirical evaluations are fine.

---

> ### Author Response · Authors · 2026-03-29
>
> We thank the reviewer for the thoughtful feedback and for recognizing the value of addressing an underexplored but practically important problem. We agree that our work can be viewed as filling a theoretical gap for a widely used heuristic, and we clarify below both the scope of our contributions and how we have strengthened the manuscript in response to the suggestions.
>
> ---
>
> ### On the nature of the theoretical contribution
>
> We agree that some components of the analysis build on classical tools such as Weyl’s inequality and incremental subspace arguments. However, we would like to emphasize that the main contribution is not the individual inequalities themselves, but their **systematic adaptation to the concatenated setting** and, crucially, their use to derive **merge certificates for clustering under explicit error constraints**.
>
> In particular, the key novelty lies in:
> - formulating **error-controlled clustering** of matrices via low-rank approximation,
> - deriving **computable upper bounds** on the reconstruction error of concatenated matrices without forming them explicitly,
> - and translating these bounds into **algorithmic procedures** that guide clustering decisions.
>
> To better highlight this perspective, we have revised the manuscript to more clearly connect the theoretical results to the clustering algorithms and to emphasize the role of the bounds as *decision rules* rather than standalone inequalities.
>
> ---
>
> ### (1) Norms other than Frobenius
>
> We thank the reviewer for this suggestion.
>
> While truncated SVD is also optimal in spectral norm, the corresponding error is determined by the $(r+1)$-th singular value and reflects only the largest residual direction. In contrast, our framework is built around controlling the **total reconstruction error**, which is naturally expressed via the Frobenius norm as cumulative discarded spectral energy.
>
> This distinction is essential for clustering: the proposed merge criteria rely on **additive energy-based certificates** that aggregate contributions across multiple directions. Such certificates do not directly translate to spectral-norm guarantees, since the spectral norm is not additive and is sensitive only to the worst-case direction.
>
> Extending the framework to spectral norm would therefore require fundamentally different merge criteria that directly control the $(r+1)$-th singular value of concatenated matrices. This represents a different objective and would require new theoretical developments.
>
> We have added a discussion in the revised manuscript to clarify this distinction and outline spectral-norm extensions as a direction for future work.
>
> ---
>
> ### (2) Faster algorithms with end-to-end guarantees / sketching
>
> We thank the reviewer for this suggestion.
>
> The approximate incremental estimator used in our paper is indeed related in spirit to streaming and sketch-based low-rank approximation methods. However, it is designed as a **deterministic plug-in estimator** and therefore does not provide end-to-end guarantees.
>
> Randomized sketching methods (e.g., randomized SVD or range finders) could be incorporated into our framework by replacing the approximate singular-value estimator with a sketch-based approximation of the dominant subspace of each candidate cluster. In this case, the merge criterion would be computed using sketched singular values, yielding error estimates that are accurate up to a $(1+\varepsilon)$ factor with high probability.
>
> This would result in a clustering procedure with **probabilistic end-to-end guarantees**, in contrast to the deterministic guarantees provided by the residual-based method.
>
> However, an important distinction remains: our framework relies on **conservative merge certificates** to ensure that the prescribed error tolerance is not violated. Standard randomized SVD guarantees approximation of the optimal rank-$r$ error, but does not directly provide **upper bounds on residual energy** required for safe merge decisions. As a result, additional care would be needed to control potential underestimation and ensure robustness of clustering decisions under randomness.
>
> We have added a discussion of this point in the revised manuscript, clarifying that sketch-based variants are a promising direction for obtaining faster algorithms with high-probability guarantees, while the present work focuses on deterministic certificates and their practical approximations.
>
> ---
>
> We thank the reviewer again for the constructive suggestions, which helped us better position the theoretical contributions and clarify possible extensions.

---

### Review · Reviewer_FQMf · 2026-03-19

**Summary Of Contributions:**

The authors consider the problem of computing singular value decompositions (SVDs) on concatenated matrices, which can be viewed as higher-order tensors that have been flattened. This is often done without consideration of the approximation error induced by the concatenation, and as far as I'm aware, there are no prior theoretical results on the approximation error as a function of the reduced representation size. The authors propose multiple approaches for bounding the relative Frobenius norm approximation error when concatenating two matrices. This forms a sort of clustering approach that determines which matrices to concatenate in order to maintain an accuracy guarantee. They demonstrate that an approximation to one of the bounds achieves strong compression at the expense of increased wall clock time.


## Strengths
- Addresses a problem that is likely of interest to a large portion of the ML audience, as low-rank representations of concatenated matrices have many applications.
- The authors provide a good combination of theoretical results (bounds on approximation error) backed up by experimental results suggesting strong performance for the approximate methods.
- Very well written with a good division of material in the main paper and appendices. (I read through the whole paper carefully but only skimmed through the appendices.)

## Weaknesses
- The target rank is a crucial hyperparameter in the proposed approaches, and the authors don't provide any insight on how to tune it. This is understandable for a more theoretically-focused paper, yet somewhat disappointing because the behavior is non-monotonic, suggesting that this target rank may be difficult to choose.
- No evaluation of the low-rank representations on downstream tasks. While I would not expect any sort of guarantees for downstream task accuracy, some experiment results could improve the paper.

**Additional Comments:**

- Is Frobenius norm the right norm to care about? Perhaps consider adding a discussion on different norms for measuring error.
- Fix references in Section 5, first paragraph, to have author names in parentheses (\citep rather than \citet), e.g., Qualcomm AI Research (2025) -> (Qualcomm AI Research, 2025).

**Audience:**

Yes

**Audience Explanation:**

Yes, this could appeal to a wide ML audience given how frequently SVDs on concatenated matrices tend to be used.

**Claims And Evidence:**

Yes

**Claims Explanation:**

Yes, both with theorems (with proofs in the appendices) and experimental results.

**Requested Changes:**

I would consider 1-3 to be critical changes and 4-5 to strengthen the paper.
1. Since the estimated $\tilde{\mathcal{E}}_r(M_K)$ is not an actual bound, do you find any instances where it underestimates error, in which case the slack $\Delta$ might actually be negative? Either way, this should be added as a comment in the experiment results.
2. How does the slack for the approximate methods behave for PDEBench, given the large increase in compression ratio for the approximate methods compared to the exact ones?
3. Are the compression and relative error entries in Table 3 directly comparable to those of Table 2, in which your approximate method with residual-based clustering achieves 2.3x compression for 0.05 relative error? If so, perhaps add this as another row in Table 3 to make a direct comparison.
4. Is there any practical utility for the exact residual-based method? It seems to achieve about the same amount of compression as the max-norm method but with significant increase in computation time.
5. The comparisons for wall clock time are all against the fast max norm method. Is there some other type of method you could compare to as a baseline? For example, a suitable higher-order SVD that provides a comparison for not concatenating matrices at all?

---

> ### Author Response · Authors · 2026-03-29
>
> We thank the reviewer for the careful reading of the paper, the positive assessment, and the constructive feedback. Below we address all concerns in the order raised.
>
> ---
>
> ## Weaknesses
>
> ### (1) Target rank selection and non-monotonicity
>
> We agree that the target rank $r$ is a crucial hyperparameter and have added a discussion in the revised manuscript.
>
> In our framework, $r$ controls truncation error and determines which merges satisfy the error constraint, making the overall compression **non-monotonic in** $r$ due to its interaction with clustering.
>
> Principled rank selection is challenging, as it depends on both approximation quality and the induced clustering structure. Our approach avoids computing singular values of concatenated matrices and instead relies on the same **surrogate error estimators** used for clustering. In practice, we treat $r$ as a hyperparameter and evaluate a small grid of candidate values, selecting the best configuration under the prescribed error constraint.
>
> ---
>
> ### (2) Downstream evaluation
>
> We thank the reviewer for this suggestion. We have added a **downstream experiment on PDEBench** (Appendix G).
> We consider a time-series forecasting task (20-step prediction) and compare no compression, full concatenation + SVD, and our clustering-based method.
>
> Key observations:
> - **Low-error regime:** performance matches the uncompressed baseline even at substantial compression (up to $\sim\times 6$).
> - **Threshold behavior:** performance degrades sharply beyond a certain error level.
> - **Naive concatenation fails:** full concatenation + SVD yields strong compression but poor downstream performance, highlighting the importance of clustering.
>
> ---
>
> ## Requested Changes
>
> ### (1) Can approximate slack be negative?
>
> Yes, in principle it can.
>
> We have:
> - added **Remark 3** in the main text,
> - included a **formal construction in Appendix D.3**,
> - and provided **Appendix F** with slack distributions under the same settings as Table 2.
>
> The key mechanism is that the approximate estimator is **order-dependent**, so truncation may discard directions that later reappear. Empirically, however, we did **not observe negative slack** across all datasets.
>
> ---
>
> ### (2) PDEBench slack behavior
>
> We added **Appendix F** with slack diagnostics for PDEBench.
>
> The large compression gains arise because exact methods are **overly conservative** on this dataset, while the approximate estimator better captures shared structure. Thus, the improvement is not due to systematic underestimation, but rather due to **reduced conservativeness**.
>
> ---
>
> ### (3) Table 2 vs Table 3 comparability
>
> Thank you for the suggestion. We have added the corresponding row.
>
> ---
>
> ### (4) Practical utility of the exact residual-based method
>
> We thank the reviewer for this insightful question.
>
> The exact residual-based method is not intended as a default, but serves two roles: (i) it provides a **tight deterministic guarantee** by accumulating all novel directions, enabling merges that max-norm rejects, and (ii) it serves as a **reference** for approximate methods, which may underestimate error (Appendix D.3).
>
> We acknowledge its higher cost and position it as a **guaranteed but slower method**, complementing max-norm (fast, conservative) and approximate methods (fast, high compression, not guaranteed).
>
> ---
>
> ### (5) Additional runtime baseline
>
> We thank the reviewer for this suggestion.
>
> We clarify that **Table 2 reports clustering time only**, i.e., the time required to determine admissible clusters under the error constraint, excluding the cost of computing SVD.
>
> Methods such as HOSVD operate on a **fixed tensor structure** and do not address clustering, and are therefore not directly comparable at this stage. More broadly, concatenated SVD learns which matrices should share a subspace, whereas tensor methods assume a predefined structure.
>
> We have clarified this distinction in the revised manuscript.
>
> ---
>
> ## Additional Comments
> ### (1) Frobenius norm vs other norms
> We thank the reviewer for this question.
>
> Frobenius norm is used because truncated SVD is optimal under it, the error equals the **sum of discarded singular-value energy**, and it enables **additive error accounting** essential for clustering.
>
> In contrast, spectral norm captures only worst-case error and is less aligned with compression.
>
> We have added a discussion clarifying this choice and possible extensions.
>
> ---
>
> ### (2) Citation formatting
>
> Thank you, corrected.
>
> ---
>
> ## Summary
>
> We have revised the manuscript to:
> - clarify rank selection and its non-monotonic behavior,
> - include downstream evaluation,
> - analyze slack behavior (including PDEBench),
> - demonstrate the possibility of negative slack,
> - improve table comparability,
> - clarify the role of the exact residual-based method,
> - and strengthen discussion of norms and baselines.
>
> We thank the reviewer again for the constructive feedback, which has helped improve both clarity and completeness of the paper.

---

> > ### Comment · Reviewer_FQMf · 2026-04-19
> >
> > Thank you for the detailed response and careful revision. I believe that this paper has been much improved.
> >
> > For future reference, it would be helpful to mark the revised portions using a different color of text.

---

### Author Response · Authors · 2026-06-02

I would like to thank the reviewers and the Action Editor for the constructive feedback and helpful suggestions throughout the review process. I appreciate the time and effort invested in evaluating this work.

---

### Decision · Action_Editor_eX6t · 2026-05-04

**Recommendation:** Accept as is

**Audience:**

Yes

**Audience Explanation:**

The results are relatively interesting, yet the algorithm is only effective in a narrow range of scenarios.

**Claims And Evidence:**

Yes

**Claims Explanation:**

The paper contains bother empirical arguments and applications:
* The major portion of the paper is to prove the Weyl-based bound and residual-based clustering bound gives desired error guarantees, with proofs.
* Experiments are also performed to show the tradeoff of these two approaches.